# RECONSTRUCTING TRAINING DATA FROM REAL WORLD MODELS TRAINED WITH TRANSFER LEARNING

## ABSTRACT

Current methods for reconstructing training data from trained classifiers are restricted to very small models, limited training set sizes, and low-resolution images. Such restrictions hinder their applicability to real-world scenarios. In this paper, we present a novel approach enabling data reconstruction in realistic settings for models trained on high-resolution images. Our method adapts the reconstruction scheme of Haim et al. (2022) to real-world scenarios – specifically, targeting models trained via transfer learning over image embeddings of large pre-trained models like DINO-ViT and CLIP. Our work employs data reconstruction in the embedding space rather than in the image space, showcasing its applicability beyond visual data. Moreover, we introduce a novel clustering-based method to identify good reconstructions from thousands of candidates. This significantly improves on previous works that relied on knowledge of the training set to identify good reconstructed images. Our findings shed light on a potential privacy risk for data leakage from models trained using transfer learning.

## 1 INTRODUCTION

Understanding when training data can be reconstructed from trained neural networks is an intriguing question that attracted significant interest in recent years. Successful reconstruction of training samples has been demonstrated for both generative models (Carlini et al., 2021; 2023) and classification settings (Haim et al., 2022). Exploring this question may help understand the extent to which neural networks memorize training data and their vulnerability to privacy attacks and data leakage.

Existing results on training data reconstruction from neural network classifiers focus on restricted and unrealistic settings. These methods require very small training datasets, which strongly limit their ability to generalize. Additionally, they are constrained to low-resolution images, such as CIFAR or MNIST images, and simple models like multilayered perceptrons (MLPs) or small CNNs.

We aim to overcome these limitations in a transfer-learning setting. Transfer Learning leverages knowledge gained from solving one problem to address a related problem, often by transferring learned representations from large pre-trained models (known as *Foundation Models*) to tasks with limited training data. In the context of deep learning, transfer learning is commonly implemented by fine-tuning the final layers of pre-trained models or training small MLPs on their output embeddings, known as deep features (Oquab et al., 2014). This approach often achieves high generalization even for learning tasks with small training sets, while also requiring less computing power. Thus, transfer learning is very common in practice.

In this work, we demonstrate reconstruction of training samples in more realistic scenarios. Specifically, we reconstruct high-resolution images from models that achieve good test performance, within a transfer learning framework. Our approach involves training an MLP on the embeddings of common pre-trained transformer-based foundation models, such as CLIP (Radford et al., 2021) or DINO-ViT (Caron et al., 2021) (see Fig. 1). Our findings have implications for privacy, particularly when transfer learning is being used on sensitive training data, such as medical data. Consequently, preventing data leakage in transfer learning necessitates the development of appropriate defenses.

Additionally, our work addresses a key limitation of prior reconstruction works: their reliance on training images for identifying good reconstructions from thousands of candidates. While this approach demonstrated that training images are embedded within the model's parameters, it's

Figure 1: Reconstructed Data from a binary classifier trained on 100 DINO-VIT embeddings

unrealistic for attackers to have access to the training data. To overcome this, we introduce a novel clustering-based approach to effectively identify reconstructed training samples, eliminating the need for prior knowledge of the training set. This marks a significant step towards establishing reconstruction techniques as real-world privacy attacks.

**Our Contributions:**

- We demonstrate reconstruction of high-resolution training images from models trained in a transfer learning approach, a significant advancement from previous reconstruction methods that were limited to small images and models with low generalization.

- We demonstrate, for the first time, reconstruction of non-visual data (feature vectors of intermediate layers).

- We introduce a novel clustering-based approach for effectively identifying training samples without a-priori knowledge of training images, a significant step towards a more realistic privacy attack.

## 2 PRIOR WORK

**Data Reconstruction Attacks.** Reconstruction attacks attempt to recover the data samples on which a model is trained, posing a serious threat to privacy. Earlier examples of such attacks include activation maximization (model-inversion) (Fredrikson et al., 2015; Yang et al., 2019), although they are limited to only a few samples per class or assume knowledge of all-but-one sample (Balle et al., 2022). Reconstruction in a federated learning setup (Zhu et al., 2019; He et al., 2019; Hitaj et al., 2017; Geiping et al., 2020; Huang et al., 2021; Wen et al., 2022) where the attacker assumes knowledge of samples' gradients. Other works studied reconstruction attacks on generative models like LLMs (Carlini et al., 2019; 2021; Nasr et al., 2023) and diffusion-based image generators (Somepalli et al., 2022; Carlini et al., 2023). Our work is based on the reconstruction method from Haim et al. (2022), which relies only on knowledge of the parameters of the trained model, and is based on theoretical results of the implicit bias in neural networks (Lyu & Li, 2019; Ji & Telgarsky, 2020). This work was generalized to multi-class setting (Buzaglo et al., 2023) and to the NTK regime (Loo et al., 2023).

**Transfer Learning.** Deep transfer learning, a common technique across various tasks (see surveys: (Tan et al., 2018; Zhuang et al., 2020; Iman et al., 2023)), leverages pre-trained models from large datasets to address challenges faced by smaller, domain-specific datasets (e.g., in the medical domain (Kim et al., 2022)). While convolutional neural networks (CNNs) have been the go-to approach for transfer learning (Oquab et al., 2014; Yosinski et al., 2014), recent research suggests that vision transformers (ViTs) may offer stronger learned representations for downstream tasks (Caron et al., 2021; He et al., 2022). For example, ViT (Dosovitskiy et al., 2020), pre-trained on ImageNet (Deng et al., 2009), provides robust general visual features. Beyond supervised pre-training, self-supervised learning methods like DINO (Caron et al., 2021; Oquab et al., 2023) learn informative image representations without requiring labeled data, allowing the model to capture strong image features suitable for further downstream tasks. Additionally, CLIP (Radford et al., 2021) has emerged as a powerful technique, leveraging a massive dataset of paired text-image examples and contrastive loss to learn semantically meaningful image representations.

Figure 2: Overview of our training and data reconstruction scheme.

## 3 METHOD

Our goal is to reconstruct training samples (images) from a classifier that was trained on the corresponding embedding vectors of a large pre-trained model in a transfer learning manner.

The classifier training is illustrated in Fig. 2a. Formally, given an image classification task $D_s = \{(\mathbf{s}_i, y_i)\}_{i=1}^n \subseteq \mathbb{R}^{d_s} \times \{1, \dots, C\}$, where $d_s$ is the dimension of the input image[1] and $C$ is the number of classes, we employ a large pre-trained model $\mathcal{F} : \mathbb{R}^{d_s} \to \mathbb{R}^d$ (e.g., DINO) to transfer each image $\mathbf{s}_i$ to its corresponding deep feature embedding $\mathbf{x}_i = \mathcal{F}(\mathbf{s}_i) \in \mathbb{R}^d$, where $d$ is the dimension of the feature embedding vector (the output of $\mathcal{F}$). We then train a model $\phi(\cdot, \boldsymbol{\theta}) : \mathbb{R}^d \to \mathbb{R}^C$ to classify the embedding dataset $D_x = (\mathbf{x}_i, y_i)_{i=1}^n \subseteq \mathbb{R}^d \times \{1, \dots, C\}$, where $\boldsymbol{\theta} \in \mathbb{R}^p$ is a vectorization of the trained parameters. Typically, $\phi$ is a single hidden-layer multilayer perceptron (MLP). Also note that $\mathcal{F}$ is kept fixed during the training of $\phi$. The overall trained image classifier is $\phi(\mathcal{F}(\mathbf{s}))$.

Our reconstruction approach is illustrated in Fig. 2b and presented in detail below. Given the trained classifier $\phi$ and the pre-trained model $\mathcal{F}$, our goal is to reconstruct training samples $\mathbf{s}_i$ from the training set $D_s$. The reconstruction scheme comprises two parts:

1. Reconstructing embedding vectors from the training set of the classifier $\phi$.

2. Mapping the reconstructed embedding vectors back into the image domain. Namely, computing $\mathcal{F}^{-1}$ (e.g., by "inverting" the pre-trained model $\mathcal{F}$).

### 3.1 RECONSTRUCTING EMBEDDING VECTORS FROM $\phi$

Given a classifier $\phi : \mathbb{R}^d \to \mathbb{R}^c$ trained on an embedding training-set $D_x = \{(\mathbf{x}_i, y_i)\}_{i=1}^n$, we apply the reconstruction scheme of (Haim et al., 2022; Buzaglo et al., 2023) to obtain $\{\hat{\mathbf{x}}_i\}_{i=1}^m$, which are $m$ "candidates" for reconstructed samples from the original training set $D_x$. In this section we provide a brief overview of the reconstruction scheme of (Haim et al., 2022; Buzaglo et al., 2023) (for elaboration see Sec. 3 in Haim et al. (2022)):

**Implicit Bias of Gradient Flow:** Lyu & Li (2019); Ji & Telgarsky (2020) show that given a homogeneous[2] neural network $\phi(\cdot, \boldsymbol{\theta})$ trained using gradient flow with a binary cross-entropy loss on a binary classification dataset $\{(\mathbf{x}_i, y_i)\}_{i=1}^n \subseteq \mathbb{R}^d \times \{\pm 1\}$, its parameters $\boldsymbol{\theta}$ converge to a KKT point of the maximum margin problem. In particular, there exist $\lambda_i \geq 0$ for every $i \in [n]$ such that the parameters of the trained network $\boldsymbol{\theta}$ satisfy the following equation:

---

[1]Typically $d_s = 3 \times h \times w$, where $h$ and $w$ are the height and width of the image, respectively.
[2]W.r.t the parameters $\boldsymbol{\theta}$. Namely $\forall c > 0 : \phi(\cdot, c\boldsymbol{\theta}) = c^L \phi(\cdot, \boldsymbol{\theta})$ for some $L$.

$$\boldsymbol{\theta} = \sum_{i=1}^{n} \lambda_i y_i \nabla_{\boldsymbol{\theta}}(\phi(\mathbf{x}_i, \boldsymbol{\theta})) \ . \tag{1}$$

**Data Reconstruction Scheme:** Given such a trained model $\phi$ with trained (and fixed) parameters $\boldsymbol{\theta}$, the crux of the reconstruction scheme is to find a set of $\{\mathbf{x}_i, \lambda_i, y_i\}$ that satisfy Eq. (1). This is done by minimizing the following loss function:

$$L_{\text{rec}}(\hat{\mathbf{x}}_1, \ldots, \hat{\mathbf{x}}_m, \lambda_1, \ldots, \lambda_m) := \left\| \boldsymbol{\theta} - \sum_{i=1}^{m} \lambda_i y_i \nabla_{\boldsymbol{\theta}}(\phi(\hat{\mathbf{x}}_i, \boldsymbol{\theta})) \right\|_2^2 , \tag{2}$$

Where the optimization variables $\{\hat{\mathbf{x}}_i, \lambda_i\}$ are initialized at random from $\lambda_i \sim \mathcal{U}(0,1)$ and $\hat{\mathbf{x}}_i \sim \mathcal{N}(0, \sigma)$ ($\sigma$ is a hyperparameter). This generates $m$ vectors $\{\hat{\mathbf{x}}_i\}_{i=1}^{m}$ that we consider as "candidates" for reconstructed samples from the original training set of the classifier $\phi$. The number of candidates $m$ should be "large enough" (e.g., $m \geq 2n$, and see discussion in Haim et al. (2022)). The $y_i$ are assigned in a balanced manner (i.e., $y_1, \ldots, y_{m/2} = 1$ and $y_{1+m/2}, \ldots, y_m = -1$). Lastly, Buzaglo et al. (2023) extended this scheme to multi-class classification problems.

The data reconstruction scheme is conducted multiple times for different choices of hyperparameters (e.g., learning rate and $\sigma$). For each trained model, we run about 50-100 reconstruction runs with $m = 500$, resulting in about 25k-50k candidates. See Appendix B.2 for full details.

## 3.2 Mapping Embedding Vectors $\hat{\mathbf{x}}_i$ to the Image Domain $\hat{\mathbf{s}}_i$

Unlike previous works on data reconstruction that directly reconstruct training images, our method reconstructs embedding vectors. To evaluate the effectiveness of our reconstructed candidates, we must first map them back to the image domain. In this section we describe how we achieve training *images* from image-*embeddings*. Namely, given reconstructed image-embeddings $\hat{\mathbf{x}}_i$, our goal is to compute $\hat{\mathbf{s}}_i = \mathcal{F}^{-1}(\hat{\mathbf{x}}_i)$. To this end we apply model-inversion methods and in particular, the method proposed in Tumanyan et al. (2022).

Given a vector $\hat{\mathbf{x}}_i$ (an output candidate from the reconstruction optimization in Section 3.1), we search for an input image $\hat{\mathbf{s}}_i$ to $\mathcal{F}$ that maximizes the cosine-similarity between $\mathcal{F}(\hat{\mathbf{s}}_i)$ and $\hat{\mathbf{x}}_i$. Formally:

$$\hat{\mathbf{s}}_i = \mathcal{F}^{-1}(\hat{\mathbf{x}}_i) = \underset{\nu}{\text{argmax}} \frac{\mathcal{F}(\nu) \cdot \hat{\mathbf{x}}_i}{\|\mathcal{F}(\nu)\| \|\hat{\mathbf{x}}_i\|} \ . \tag{3}$$

We further apply a Deep-Image Prior (DIP) (Ulyanov et al., 2018) to the input of $\mathcal{F}$. I.e., $\nu = g(\mathbf{z})$ where $g$ is a CNN U-Net model applied to a random input $\mathbf{z}$ sampled from Gaussian distribution. The only optimization variables of the inversion method are the parameters of $g$. See Appendix B.3 further explanation and full implementation details.

By applying model-inversion to DINO embeddings, Tumanyan et al. (2022) demonstrated that the [CLS] token contains a significant amount of information about the visual appearance of the original image from which it was computed. Even though their work was done in the context of image to image style transfer, their results inspired our work and motivated us to apply their approach in the context of reconstructing training image samples.

A significant modification to Tumanyan et al. (2022) in our work is by employing a cosine-similarity loss instead of their proposed MSE loss. We find that using MSE loss (i.e., $\mathcal{F}^{-1}(\hat{\mathbf{x}}) = \text{argmin}_\nu \|\mathcal{F}(\nu) - \hat{\mathbf{x}}_i\|^2$) is highly sensitive to even small changes in the scale of $\hat{\mathbf{x}}$. The scales of $\hat{\mathbf{x}}$ can be very different from the unknown $\mathbf{x} = \mathcal{F}(\mathbf{s})$. Using cosine similarity alleviates this issue while simultaneously achieving similar quality for the inverted image result (see also Appendix A.1).

The above-mentioned technique is used for mapping embeddings to images for most transformers that we consider in our work. However, this technique did not produce good results when applied to CLIP (Radford et al., 2021). Therefore, to map CLIP image embeddings to the image domain, we employ a diffusion-based generator conditioned on CLIP embeddings by Lee et al. (2022) (similar in spirit to the more popular DALL-E2 (Ramesh et al., 2022); see also Appendix A.7 and Appendix B.4).

### 3.3 Selecting Reconstructed Embeddings to be Inverted

Applying the model-inversion described in Section 3.2 to a large pretrained model is computationally intensive. Inverting a single embedding vector takes about 30 minutes on an NVIDIA-V100-32GB GPU. Therefore, it is not feasible to invert all 25k-50k output candidates of Section 3.1.

To determine which reconstructed candidates to invert, we pair each training embedding $\mathbf{x}_i$ with its nearest reconstructed candidate $\hat{\mathbf{x}}_j$ (measured by cosine similarity) and select the top 40 vectors with the highest similarity for inversion. This approach proves effective in practice, yielding images with high visual similarity to the original training images, as demonstrated in the results (e.g., Fig. 1).

In practice, the original training embeddings are not available (and inverting all candidates is computationally prohibitive). In Section 5 we introduce a novel method to identify good reconstructions without relying on either ground-truth embeddings or exhaustive inversion.

## 4 Results

We demonstrate reconstructed training images from models trained in a transfer learning setup, on the embeddings of large pretrained models. We train several MLPs to solve learning tasks for various choices of training images and choices of the large pretrained backbones from which the image embeddings are computed.

**Datasets.** Since we simulate a model that is trained in a transfer learning manner, it is reasonable to assume that such tasks involve images that were not necessarily included in the training sets on which the pretrained backbone was trained (typically, ImageNet (Deng et al., 2009)). In our experiments we use images from **Food-101** (Bossard et al., 2014) (most popular dishes from foodspotting website) and **iNaturalist** (Van Horn et al., 2018) (various animals/plants species) datasets. The resolution of the images vary between 250-500 pixels, but resized and center-cropped to $224 \times 224$.

**Pretrained Backbones ($\mathcal{F}$) for Image Embeddings.** We select several Transformer-based foundation models that are popular choices for transfer learning in the visual domain:

- **ViT** (Dosovitskiy et al., 2020): vit-base-patch16-224 from TIMM Wightman (2019).
- **DINO-ViT** (Caron et al., 2021): dino-vitb16 from the official implementation[3].
- **DINOv2** (Oquab et al., 2023): dinov2-vitb14-reg from the official implementation[4].
- **CLIP-ViT** (Radford et al., 2021): ViT-L/14 as provided by OpenAI's CLIP repository [5].

The dimension of the output embeddings is consistent across all backbones $\mathcal{F}$, and equal to $d$=768.

**Multilayer Perceptron ($\phi$)** consists of a single hidden layer of dimension 500 ($d$-500-$C$) that is optimized with gradient descent for 10k epochs, weight-decay of 0.08 or 0.16 and learning rate 0.01. All models achieve zero training error.

Reconstructing Training Data from $\phi(\mathcal{F})$

We train classifiers $\phi(\mathcal{F}(s))$ on two binary classification tasks: (1) *binary iNaturalist* is fauna (bugs/snails/birds/alligators) vs. flora (fungi/flowers/trees/bush) and (2) *binary Food101* is beef-carpaccio/bruschetta/caesar-salad/churros/cup-cakes vs. edamame/gnocchi/paella/pizza/tacos. Each binary class mixes images from several classes of the original dataset (images are not mixed between different datasets). Each training set contains 100 images (50 per class). The test sets contains 1000/1687 images for iNaturalist/Food101 respectively. All models achieve test-accuracy above 95% (except for DINO on Food101 with 85%. Also see Fig. 31).

In Fig. 3 we show the results of reconstructing training samples from 8 models (for two binary tasks and 4 choices of $\mathcal{F}$). For each reconstructed image ($\hat{\mathbf{s}} = \mathcal{F}^{-1}(\hat{\mathbf{x}})$), we show the nearest image from

---

[3]`https://github.com/facebookresearch/dino`
[4]`https://github.com/facebookresearch/dinov2`
[5]`https://github.com/openai/CLIP`

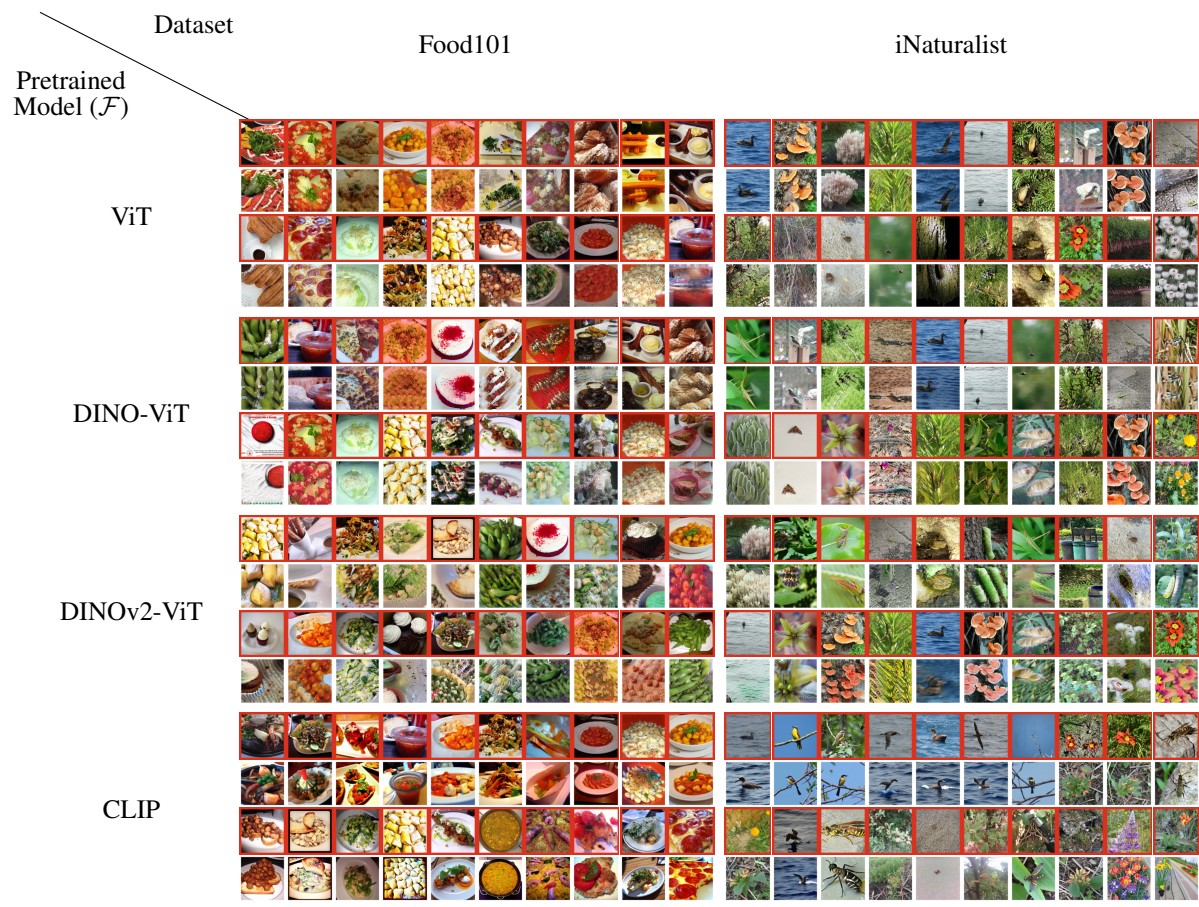

Figure 3: Training samples (red) and their best reconstructed candidate, from MLPs trained on embeddings of various backbone models for two datasets.

the training set, in terms of cosine-similarity between the embeddings of both ($d_{cosine}(\hat{\mathbf{x}}, \mathcal{F}(\mathbf{s}))$). As can be seen, many reconstructed images clearly have high semantic similarity to their corresponding nearest training images.

The quality of the results greatly depends on the effectiveness of the inversion method, which can vary across different backbones $\mathcal{F}$. DINO and ViT yield the highest quality reconstructed samples. DINOv2 proves harder to invert, resulting in lower reconstruction quality. With CLIP, we utilize UnCLIP[6] to project embeddings into good natural images, maintaining semantic similarity even as reconstruction quality decreases (e.g., same class). In Section 6 we further discuss the differences and limitations of inversion.

Our approach is also applicable to multiclass setting by using Buzaglo et al. (2023) extension of the method described in Section 3.1 (see Appendix B.5 for details). This is demonstrated in Fig. 4 where we show reconstructed training samples from models trained on multiclass tasks.

QUANTITATIVE EVALUATION OF THE RECONSTRUCTED DATA

We evaluate our results by how well they corroborate with the theory on which the reconstruction method is based, and also by how well the reconstructed images resemble the original training samples.

---

[6]We use the UnCLIP implementation from https://github.com/kakaobrain/karlo

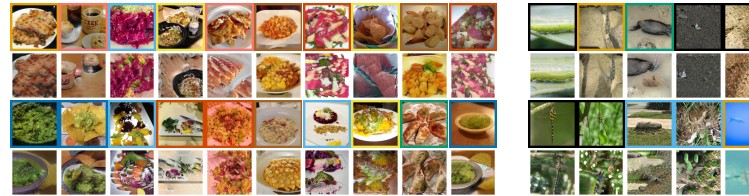

(a) DINO Food101 10 classes          (b) DINO iNaturalist 4 classes

Figure 4: Reconstructions from a multiclass models trained $100$ images from Food101/iNaturalist with $C$=$10/4$ classes ($10/25$ images per class), with test-accuracy $84\%/96\%$ (on a/b respectively). Color-padded images are training images, where color represents different classes.

**Measuring Reconstruction Quality and Alignment with Theory.** Convergence to the KKT solution of the maximum-margin implies that reconstruction is only possible for samples lying on the margin, i.e., those with the smallest model outputs.[7] This can be demonstrated by plotting each training sample's reconstruction quality (typically measured using SSIM (Wang et al., 2004)) between the original and reconstructed images), against its proximity to the decision boundary (measured by the model output).

When reconstructing images from embeddings, as in our work, the reconstructed samples may exhibit small translations or subtle artifacts that are hard to pinpoint, despite appearing visually similar. As a result, conventional image metrics like SSIM, which are sensitive to pixel alignment, may not be effective for this task.

**Quantitative Evaluation.** In Fig. 5a, we show results for several metrics for reconstruction quality, including SSIM (Wang et al., 2004), LPIPS (Zhang et al., 2018), and Split-product (Somepalli et al., 2022), as well as cosine similarity in the embedding domain ($d_{cosine}(\hat{\mathbf{x}}, \mathcal{F}(\mathbf{s}))$). Notably, cosine similarity aligns most closely with the theoretical predictions: higher values correspond to samples that are closer to the margin. In Fig. 5b we demonstrates that cosine-similarity between embeddings also aligns well with visual similarity. To this end we sort all reconstructed samples according to $d_{cosine}(\hat{\mathbf{x}}, \mathcal{F}(\mathbf{s}))$. Note how samples with high cosine-similarity also appear visually similar.

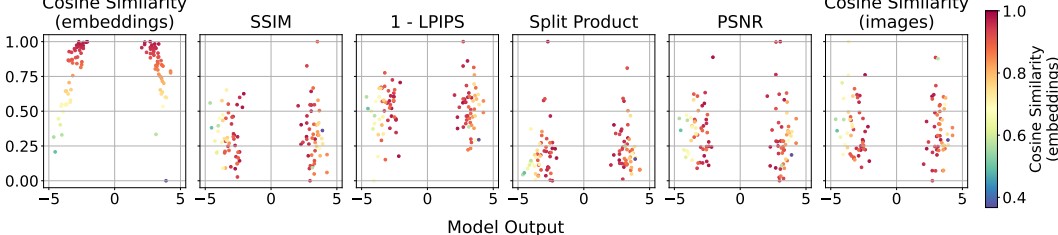

(a) Various metrics for reconstruction quality (normalized to [0,1]. I.e., $(x - \min(x))/(\max(x) - \min(x))$ where $x$ is the array containing the metric values).

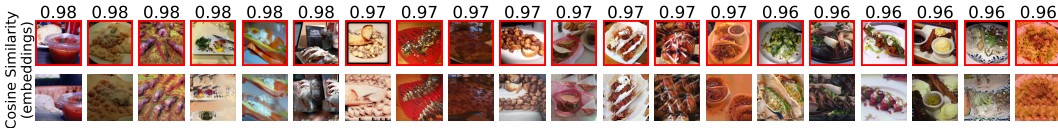

(b) Reconstructed samples sorted by $d_{cosine}(\hat{\mathbf{x}}, \mathcal{F}(\mathbf{s}))$ (values shown above images)

Figure 5: **Cosine-Similarity between embeddings (top-left) aligns well with both theoretical properties and visual similarity.** Results for DINO Food101 model. Complete results for all models and metrics are in Appendix A.2.

Such plots (reconstruction-quality vs. model-output) are a good way to summarize the reconstruction results for each model, since they show the full reconstruction quality for all samples. In Fig. 6

---

[7]In addition to the condition in Eq. (1), $\lambda_i \neq 0$ holds only for samples $\mathbf{x}_i$ that lie on the classification margin, closest to the decision boundary. See Sections 3.2 & 5.3 in Haim et al. (2022).

we show such plots for every model from Figs. 3 and 4 (where reconstruction-quality is measured by cosine similarity between embeddings). This analysis hints that samples that are closer to the classification margin (either in the binary or multiclass case) are more vulnerable to reconstruction (since their reconstruction quality is higher).

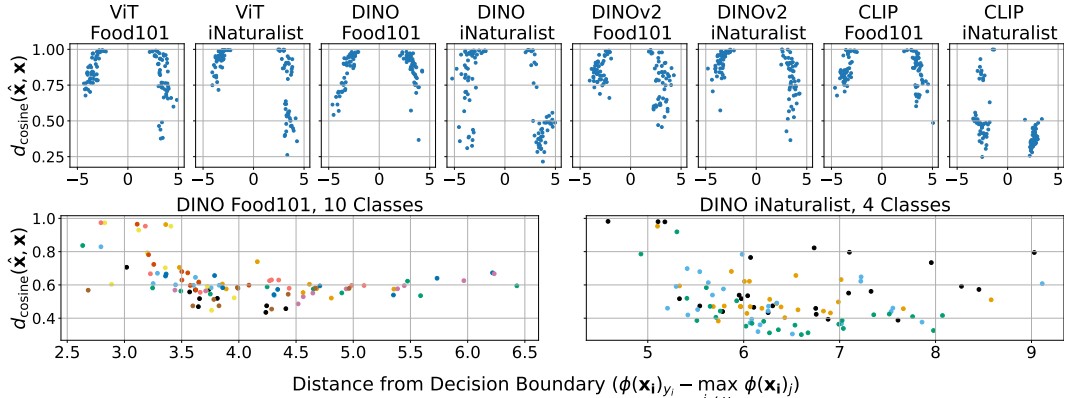

Figure 6: Quantitative summary for all models whose reconstructed samples are in Figs. 3 and 4.

## 5  IDENTIFYING GOOD RECONSTRUCTION WITHOUT THE ORIGINAL TRAINSET

In this section, we introduce a clustering-based approach to identify "good" reconstructed candidates without relying on the original training data. This is an important step towards an effective privacy attack. Previous works (Haim et al., 2022; Buzaglo et al., 2023; Loo et al., 2023), including Section 3.3 in this work, rely on the original training images for demonstrating that training images are embedded in the model parameters. However, it is not applicable to real-world privacy attacks, as attackers don't have access to the original training data.

When directly reconstructing training images, this issue can be mitigated by manual inspection of the thousands of output image candidates – a time-consuming but feasible approach. However, this approach is irrelevant when reconstructing image embeddings. The reconstructed embeddings must first be inverted into images, which is computationally expensive (inverting a single vector takes about 30 minutes on an NVIDIA-V100-32GB GPU, as detailed in Section 3.3). Inverting thousands of embeddings is simply infeasible.

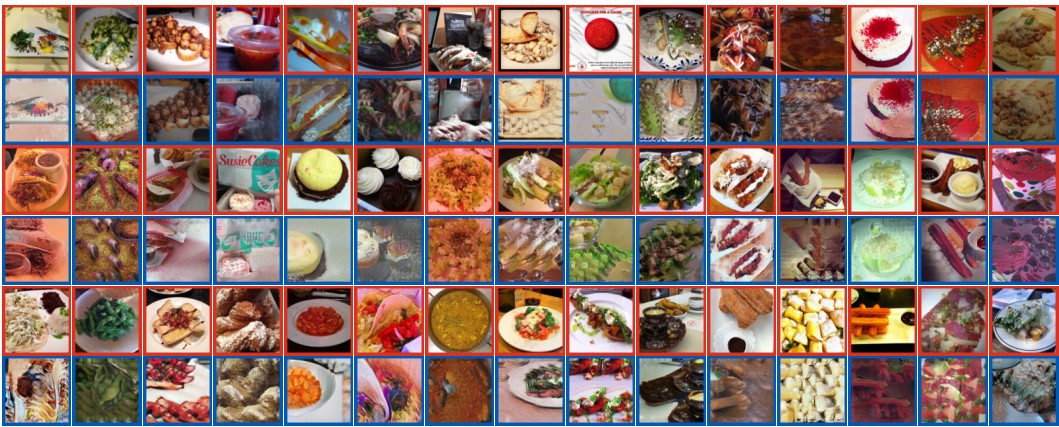

Figure 7: **Clustering-Based Reconstruction.** Inversion of clusters representatives (blue) compared to training samples whose embeddings are in the same cluster (in red).

| Pretrained Model ($\mathcal{F}$): | ViT | DINO-ViT | DINOv2-ViT | CLIP |
|---|---|---|---|---|
| **Dataset** | | | | |

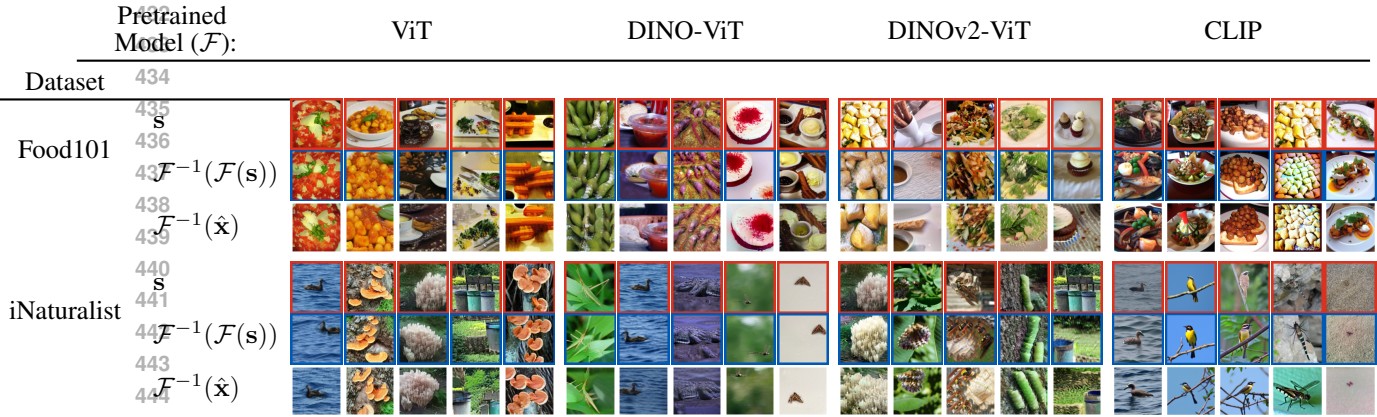

Figure 9: Training samples (red), inversion of original embeddings (blue), and inversion of reconstructed embeddings.

This is where our proposed clustering approach comes in. We observe that reconstructed candidates whose inversions are visually similar to training samples tend to cluster together. By applying clustering algorithms, we group similar candidates and only invert representative samples from the largest clusters. This reduces the total number of inversions by two orders of magnitude (from thousands to tens) and eliminates reliance on training data for identifying good reconstructed samples.

We demonstrate this by using agglomerative clustering[8] on 25,000 candidates reconstructed from a Dino-ViT-based model trained on the Food101 dataset (same as in Fig. 3). We use cosine similarity as the distance metric with "average" linkage and 1,000 clusters, from which we select the 45 largest ones (containing between 100 and 8,000 candidates each). Within each cluster, a representative is chosen by averaging all candidate embeddings. Finally, these representatives are inverted using the methods described in Section 3.2. Fig. 7 shows the results of inverting these cluster representatives (blue), along with a training sample whose embedding belongs to the same cluster (red). As can be seen, the clustering-based approach provides a very good method for reconstructing training samples without requiring the training data.

The choice of the number of clusters (MAXCLUST) significantly affect the results of our clustering-based approach. Since assessing this effect in our current image-embedding setup is computationally prohibitive, we evaluate our approach on 50k reconstructed candidates from a model trained on 500 CIFAR-10 images (same as in Haim et al. (2022)). For each MAXCLUST, we select the representatives of the largest 150 clusters by either averaging all cluster candidates (red) or selecting the nearest candidate to the cluster-mean (blue). We compare each representative to a training image in the same cluster (using SSIM) and count the numbers of good representatives (SSIM> 0.4), the results are in Fig. 8.

Notably, beyond a certain small threshold, any MAXCLUST yields a considerable amount of good reconstructed samples (see also Appendix A.8).

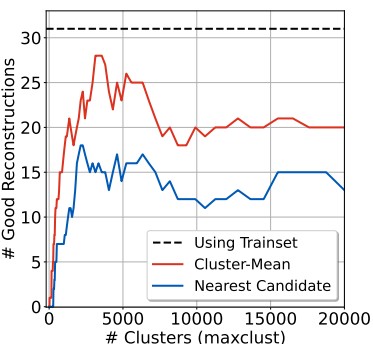

Figure 8: Impact of Num. Clusters on Reconstruction Quality (for CIFAR10 model with $n$=500)

## 6 LIMITATIONS

In this work, we made design choices when training the models to align with realistic transfer learning practices. However, some choices led to better reconstruction results than others, revealing limitations of our method. Here we discuss these limitations, their impact on our results, and identify potential future research directions:

---

[8] https://docs.scipy.org/doc/scipy/reference/generated/scipy.cluster.hierarchy.fcluster.html

- The quality of reconstructed images relies heavily on the backbone model ($\mathcal{F}$) and the inversion method (Section 3.2). Fig. 9 shows the inverted original embeddings $\mathcal{F}^{-1}(\mathcal{F}(\mathbf{s}))$ (blue), which are the "best" we can achieve (independent of our reconstruction method). It also shows how some backbones are easier or harder to invert, as evident in the difference between $\mathcal{F}^{-1}(\mathcal{F}(\mathbf{s}))$ (blue) and the original image $\mathbf{s}$ (red), for different $\mathcal{F}$'s. It can also be seen that the inverted reconstructed embeddings $\mathcal{F}^{-1}(\hat{\mathbf{x}})$ are sometimes more similar to $\mathcal{F}^{-1}(\mathcal{F}(\mathbf{s}))$ than to $\mathbf{s}$, which may hint that the challenge lies in the inversion more than in the reconstruction part. Certainly, improving model inversion techniques is likely to enhance the quality of reconstructed samples.

- CNN-based backbones $\mathcal{F}$ (e.g., VGG (Simonyan & Zisserman, 2014)) proved more challenging for inversion than Transformer-based backbones $\mathcal{F}$. Since Transformers are also being more frequently employed due to their better generalization, we decided to focus our work on them and leave CNN-based backbones for future research.

- Linear-Probing (i.e. train a single linear layer $\phi$) is common practice in transfer learning. However, current reconstruction methods, including ours, struggle to perform well on linear models. This may stem from the small number of parameters in linear models (see Appendix A.5).

- We use weight-decay regularization since it is a fairly common regularization technique. However, the reconstruction method is known to perform much worse on models that are trained without it (Buzaglo et al., 2023).

- We experimented with an embedding vector that is a concatenation of [CLS] and the average of all other output tokens (of $\mathcal{F}$). This had minor effect on the results, see Appendix A.6 for details.

- Fine-tuning the entire model $\mathcal{F}$ (together with $\phi$) is resource-intensive and less common compared to training only on fixed embedding vectors. While we followed the latter approach, full fine-tuning can be an interesting future direction.

## 7 CONCLUSION

In this work, we extend previous data reconstruction methods to more realistic transfer learning scenarios. We demonstrate that certain models trained with transfer learning are susceptible to training set reconstruction attack. Given the widespread adoption of transfer learning, our results highlight potential privacy risks. By examining the limitations of our approach, we identify simple mitigation strategies, such as employing smaller or even linear models, increasing training set size or training without weight-decay regularization. However, some of these mitigation (removing regularization or using smaller models) may also come at a cost to the generalization of the model. Furthermore, these techniques may not be effective against future advanced reconstruction attacks. We aim for our work to inspire the development of new defense methods and emphasize the importance of research on data reconstruction attacks and defenses.

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

# Appendix

## Table of Contents

## A  ADDITIONAL EXPERIMENTS

### A.1  IMPORTANCE OF COSINE-SIMILARITY FOR INVERSION (AS OPPOSED TO MSE)

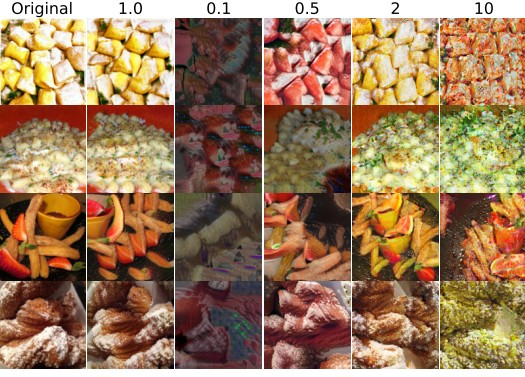

Figure 10: Inverting DINO ($\mathcal{F}^{-1}(a\mathcal{F}(\mathbf{s}))$) with different scales $a$

In Fig. 10, we illustrate the significance of having the correct scale when inverting an embedding (using the inversion described in Section 3.2). For several images $\mathbf{s}$ (left-most column), we display

the inversion of their embeddings $\mathcal{F}^{-1}(\mathcal{F}(\mathbf{s}))$ (second from left column) alongside other inversions of the same vector multiplied by varying scales, namely, $\mathcal{F}^{-1}(a\mathcal{F}(\mathbf{s}))$ for $a = \left[\frac{1}{10}, \frac{1}{2}, 2, 10\right]$. As clearly evident, inverting the same vector without knowing the "true" scale ($a = 1.0$) would result in very different results, sometimes making them hard to recognize.

The original paper Tumanyan et al. (2022) uses MSE in its inversion scheme. However, the output candidates from the reconstruction method (described in Section 3.1) can have significantly different norms than their corresponding original training embedding.

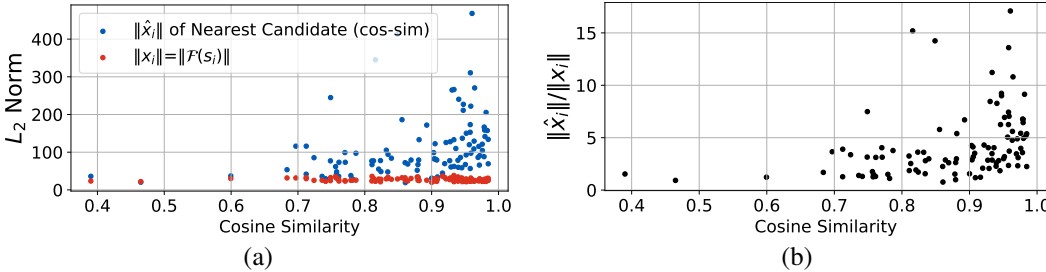

(a)            (b)

Figure 11: Comparing (a) the norms of $\mathcal{F}(\mathbf{s})$ (red) and its NN $\hat{\mathbf{x}}$ (blue), and (b) their ratios

To conduct a comparison, we employ a binary model trained on DINO embeddings of images from Food101, reconstructing candidates $\hat{\mathbf{x}}$ from this model. In Fig. 11a, for each training image $\mathbf{s}$ we compare the norm of its DINO embedding $\|\mathcal{F}(\mathbf{s})\|$ (red), to the norm of its nearest neighbour embedding $\|\hat{\mathbf{x}}\|$ (blue), where $\hat{\mathbf{x}} = \arg\min_{\mathbf{x}} d_{cosine}(x, \mathcal{F}(\mathbf{s}))$ (and the value of $d_{cosine}$ is the x-axis).

In Fig. 11b we show the ratio between the two, highlighting that candidates can have very different norm compared to their corresponding training image. This variation in norms is a result of the reconstruction scheme that we use (Section 3.1), whose nature we don't fully understand yet. However, using cosine-similarity loss in our inversion scheme eliminates this issue.

## A.2 FULL RESULTS FOR FIG. 5

In Fig. 2 we showed the results of various metrics for reconstruction-quality for a model that was trained on embeddings of DINO on Food101 dataset. We also showed alignment for visual similarity of cosine-similarity in embedding space.

In Fig. 12 we provide the full results (same as in Fig. 5a) for all other models from Fig. 3 and all metrics.

The complete results for Fig. 5b are provided in the supplementary material (sorted reconstructed sample by all 6 metrics, for each of the 8 models from Fig. 3)

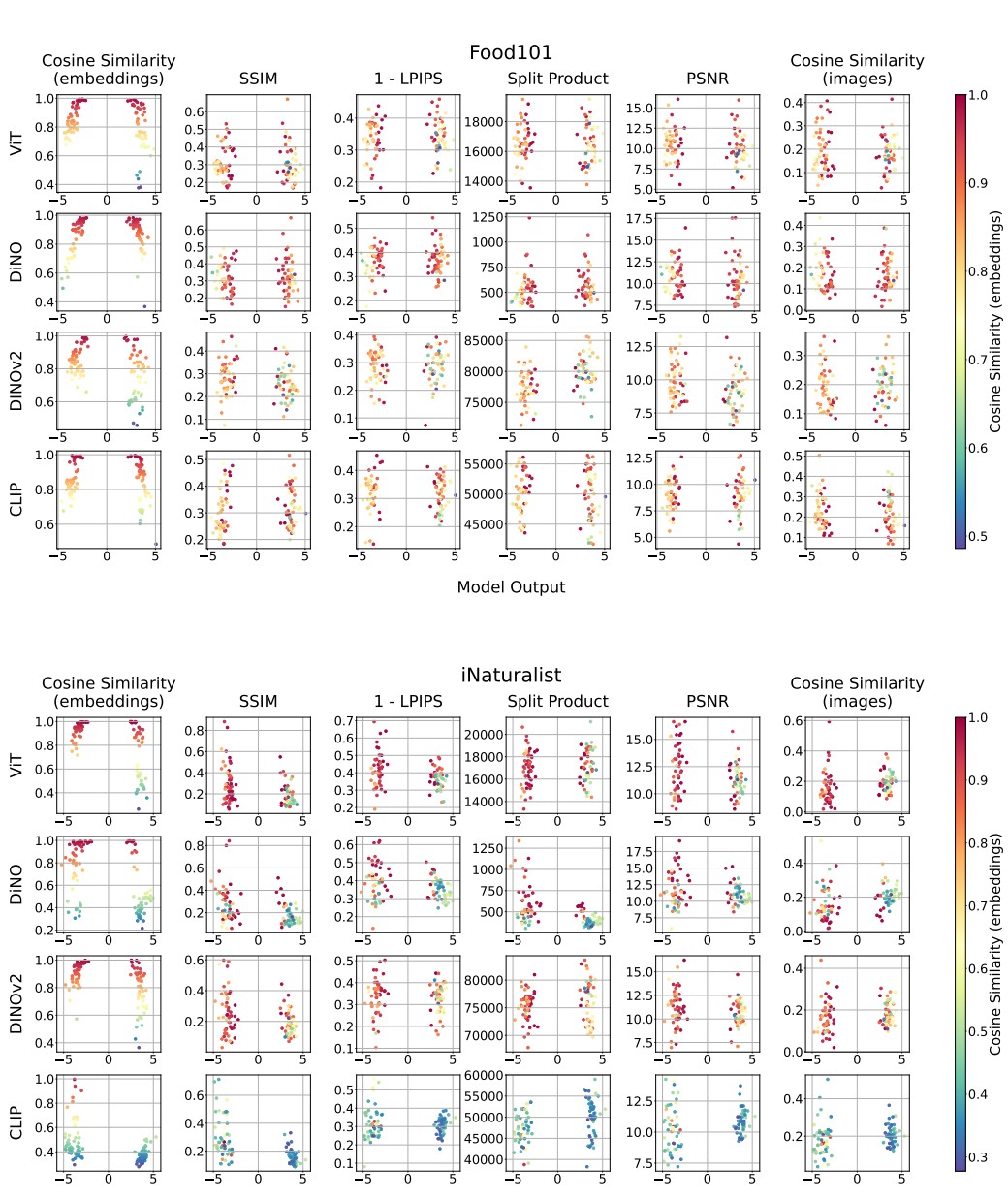

Figure 12: Various Metrics for Reconstruction-Quality vs. Model-Output

### A.3 COSINE-SIMILARITY AS A PROXY FOR GOOD RECONSTRUCTIONS

Determining whether a candidate is a reconstruction of an original sample is a difficult challenge. It is highly unlikely that the two will be exactly the same, which is why selecting an appropriate similarity measure is important. Unfortunately, there is no "best" metric for comparing two images, which is a known open problem in computer vision (see e.g., Zhang et al. (2018)).

The task becomes even more complex when dealing with reconstructed embedding vectors, as in our work. Given our computational constraints, we must choose wisely which embeddings to invert, adding another layer of complexity to the comparison process. Throughout our work, we frequently employ cosine similarity as a metric for evaluating embedding similarities. However, whether this metric accurately reflects visual quality is unclear. We set out to explore this question empirically.

Previous work on data reconstruction (Haim et al., 2022; Buzaglo et al., 2023) directly reconstruct training images, allowing us to a directly compare between cosine similarity and image similarity measures. Both works established SSIM (Wang et al., 2004) as a good visual metric for CIFAR10 images (see Appendix A.2 in Buzaglo et al. (2023)), and defined SSIM> 0.4 as a good threshold for declaring two images as sufficiently similar. In fact these works also use cosine similarity to find nearest neighbors between candidates and training images when normalized to $[-1, 1]$, and only after shifting the images to [0,1] they use SSIM. Which means that they also implicitly assume that cosine-similarity is a good proxy for visual similarity.

In Fig. 13, we quantitatively evaluate this assumption using reconstructed images from a CIFAR10-trained model (as in Haim et al. (2022)). The left panel is for simply reproducing the results of Haim et al. (2022). By looking at both middle and right panel, we see that CosSim=0.75 is a good cut-off for determining "good" reconstruction, since from this point there is a good correlation between the two metrics. This is also the reason that we use this threshold for determining good reconstruction in other experiments in the paper.

By further observing the middle panel: if SSIM> 0.4 (horizontal black line) is considered a criterion for good image reconstruction, then cosine similarity (CosSim> 0.75, vertical black line) may overlook some potentially high-quality reconstructions, indicating room for further improvement in our approach.

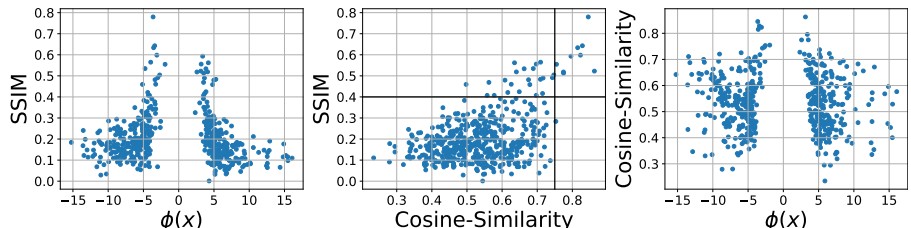

Figure 13: Comparing Cos-Sim to SSIM of training data (model trained on CIFAR10)

### A.4 DOES TRAINING DATA RECONSTRUCTABILITY REQUIRE OVERTRAINING?

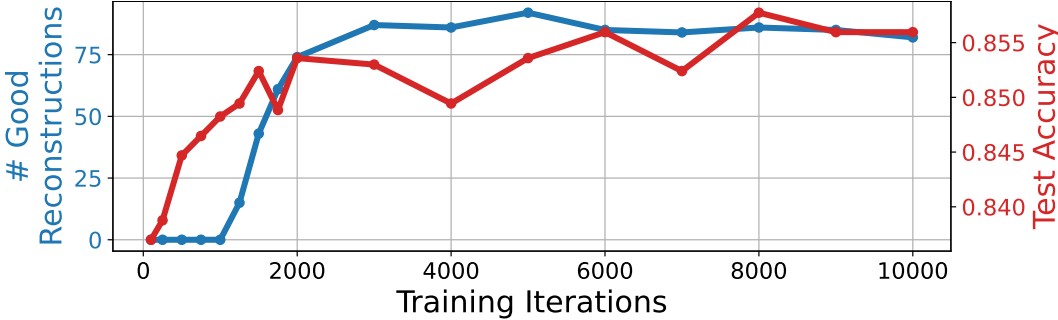

Figure 14: Does Training Data Reconstructability Require Overtraining? – Seems Not.

We set to explore how "reconstructability" (i.e., how many good samples we can reconstruct from) depends on the number of training iterations. We note from empirical observations that reconstructability certainly improves with longer training, which should not be surprising because according to theory, the model converges more to the KKT solution.

But the key question is - does the model have to be "overtrained" before becoming reconstructable, or not? To define "overtrained", we observe how the generalization accuracy increases. Obviously, the longer we train, the better the model will be reconstructable. But is it reconstructable before the generalization accuracy saturates? (Or do we have to keep training long after that?)

In Fig. 14 we show the test accuracy per training iteration (red) for a model trained on DINO embeddings from the Food101 dataset. We also show reconstruction quality (blue) by counting the number of training samples whose cosine similarity to its nearest neighbor candidate was above $0.75$. As can be seen, reconstructability increases after about 1000 iterations and starts saturating at about 2000 iterations, where the test accuracy (even though quite high in the beginning), keeps increasing by more than 1.5% until 10k iterations.

The implication is that reconstructability is achieved in a reasonable time (measured by the time taken to achieve good generalization accuracy). This observation is important to assert the realism of our method as a viable privacy threat to models trained in a similar fashion.

## A.5 IMPACT OF MODEL SIZE AND TRAINING SET SIZE ON RECONSTRUCTABILITY

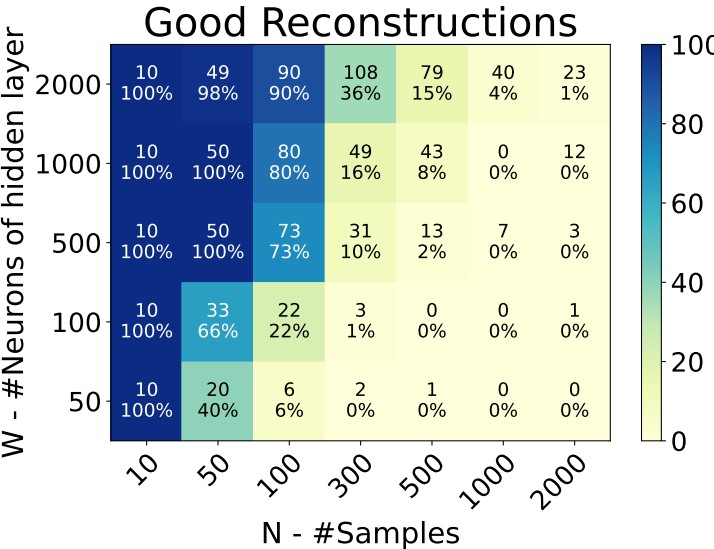

Figure 15: Effect of model size and dataset size on reconstructability.

Previous works Buzaglo et al. (2023) observed that the quality of reconstruction results is influenced by the size of the model (i.e., number of parameters) and the size of the training set. We conduct similar analysis for our models.

This relationship can be intuitively understood by considering Eq. (2) as a system of equations to be inverted, where the number of equations corresponds to the number of parameters in the model, $\theta \in \mathbb{R}^p$, and the unknowns are the coefficients $\lambda_i \in \mathbb{R}$ and the reconstructed embeddings $\mathbf{x}_i \in \mathbb{R}^d$ for each training sample $i \in \{1, ..., n\}$. The ratio $\frac{p}{n(d+1)}$ represents the number of model parameters relative to the total number of unknowns. As this ratio increases, i.e., when the model has more parameters compared to the number of unknowns, we hypothesize that the system of equations becomes more well-determined, leading to higher reconstructability.

This hypothesis is supported by the empirical results presented in Fig. 15, where we train 2-layer MLPs with architecture $D$-$W$-1 on $N$ training samples from binary Food101. Each cell reports the number of good reconstructions (cosine similarity between training embedding and its nearest

neighbor candidate > 0.75), both in absolute terms and as a percentage relative to N. As shown, when the model has more parameters relative to the number of training examples (further left and higher up in the table), our method can extract more reconstructions from the model.

This figure also show that our method can be extended to larger datasets, up to $N$=2000 (and probably beyond).

### A.6 EFFECT OF USING [CLS]+MEAN VS [CLS] AS FEATURE VECTOR

In our work we use the [CLS] token as the feature vector for a given image. However, there may be other ways to use the outputs of transformer-based foundation models as feature vectors. As suggested in Caron et al. (2021) (linear probing section), one might use a concatenation of the [CLS] token and the mean of the rest of the other output tokens ([CLS]+MEAN). In Fig. 16 we show reconstructed results for a model that was trained using such [CLS]+MEAN feature vector (using DINO on Food101). As seen, the extra information in the feature vector does not seem to have a significant effect on the total results of the reconstruction (as opposed to a possible assumption that extra information would result in higher reconstructability). While this is by no means an exhaustive evaluation of this design choice (using [CLS]+MEAN vs. just [CLS]), it does look like this may not change the results of the reconstruction too much.

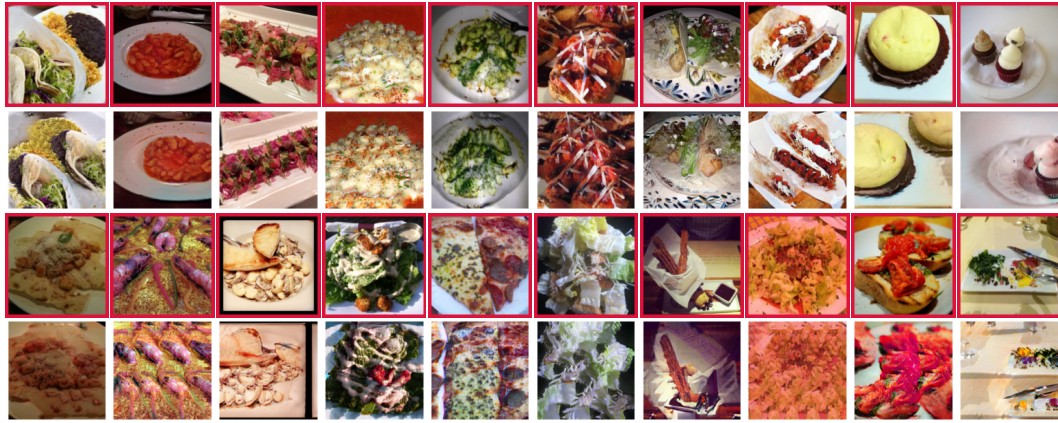

Figure 16: Reconstruction from a DINO model trained on [CLS]+MEAN embedding vector (original training image in red)

### A.7 MODEL INVERSION FOR CLIP VS UNCLIP DECODER

As described in Section 3.2, for inverting CLIP embeddings, we use an UnCLIP decoder instead of the model inversion approach used for other backbone models (ViT/DINO/DINOv2). The main reason behind this choice is that the same inversion method did not seem to provide satisfactory results for CLIP. In Fig. 17, we show output images of inverted embeddings using the approach from Tumanyan et al. (2022) (with the modifications described in our paper). The results do not produce comparable quality to using the UnCLIP decoder.

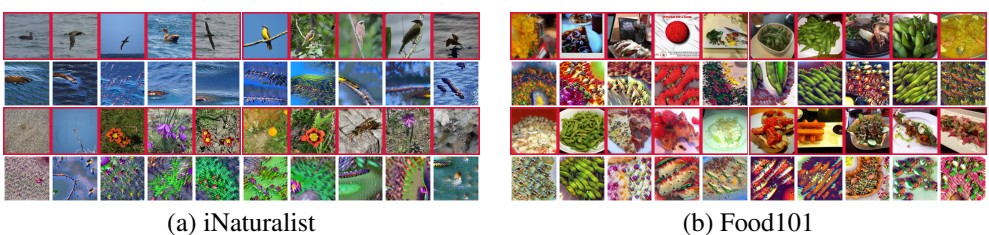

(a) iNaturalist          (b) Food101

Figure 17: Model-Inversion reconstructions from a model trained on CLIP embeddings

## A.8   FURTHER INSIGHTS ON CLUSTERING-BASED RECONSTRUCTION (SECTION 5)

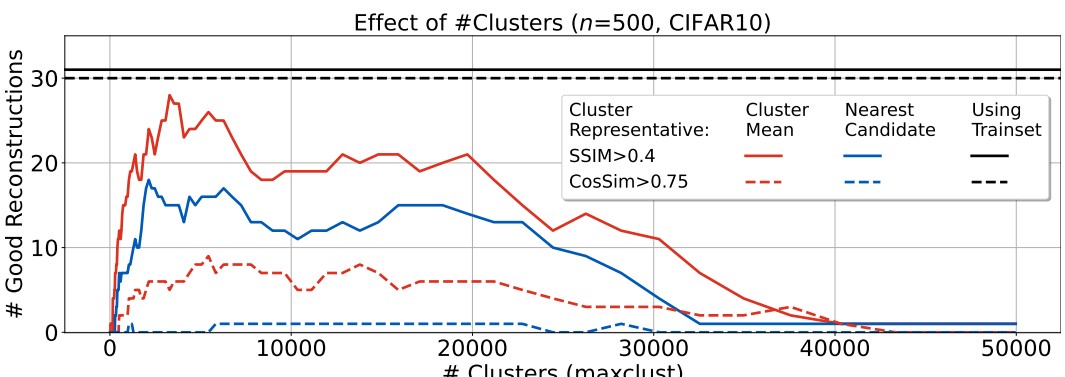

Figure 18: Extended Results for the figure in Section 5

In Fig. 18, we show extended results of the inset Figure in Section 5, displaying the same graph up to larger MAXCLUST values (red and blue solid lines), together with similar results that count the number of "good" reconstructions with CosSim $> 0.75$ (dashed blue and red lines).

The reason for the decrease in the number of good reconstructions as the number of clusters increases, is that we only consider the largest 150 clusters (per partition of the candidates, as determined by MAXCLUST). Consequently, when there are too many clusters, the probability that the largest ones correspond to a cluster of a training sample decreases (for 50k clusters, this becomes totally random). Note that the largest number of clusters in the graph is slightly smaller than 50k, and there exist several clusters with 2-3 candidates.

Another insight from this graph is that averaging several candidates together results in better candidates, an observation also made by Haim et al. (2022). In our work, we don't use such candidate averaging (except for the clustering experiments), but this may lead to improved results. We leave this for future research.

We note that since the similarity measure between candidates is cosine similarity, this implicitly applies a spherical topography for comparing candidates. Therefore, it is not straightforward to compute the mean of several candidates. In our work, we use the simple arithmetic mean, which empirically seems to work well. We considered computing the Fréchet mean, i.e., the mean of the candidates that lies on the sphere, but could not find a working implementation for this. This may also be an interesting direction for future research.

For completeness, we show how the reconstructed samples look for the choice of the "peak" SSIM from Fig. 18, which occurs at 3294 clusters. These are shown in Fig. 19.

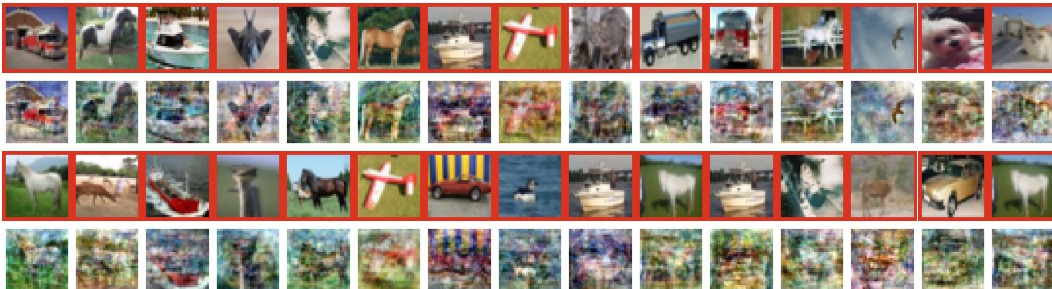

Figure 19: Reconstructed candidates of CIFAR10 model, obtained with our clustering-based approach for the "peak" value in Fig. 18 (3294)

## A.9 MORE CLUSTERING-BASED RECONSTRUCTION RESULTS

In Fig. 20 we more results of our clustering-based approach, in addition to the results in Fig. 7 (for a model trained on DINO embeddings of Food101 images).

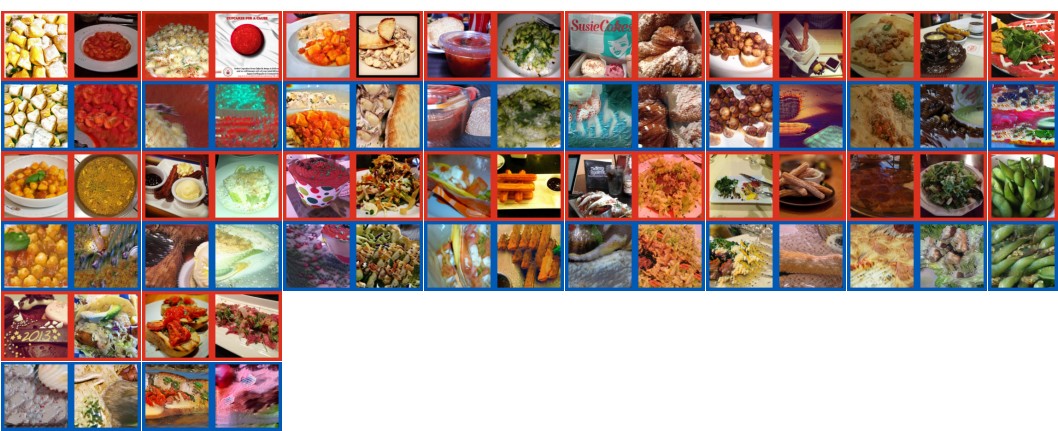

(a) ViT embeddings of Food101 images

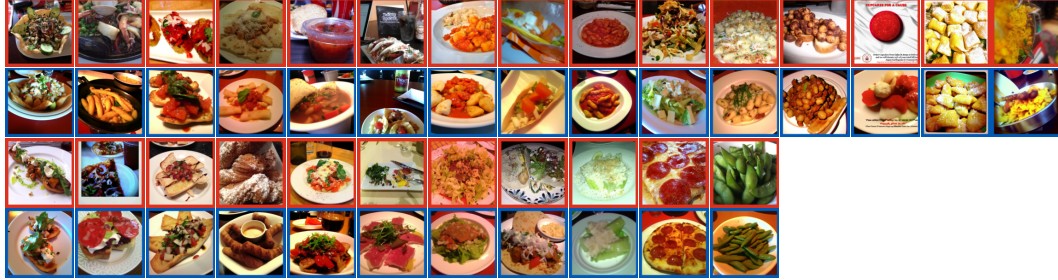

(b) CLIP embeddings of Food101 images

Figure 20: Clustering-based Reconstruction for models trained on (a) ViT embeddings of Food101 images; (b) CLIP embeddings of Food101 images

## A.10 COMPARISON TO ACTIVATION MAXIMIZATION

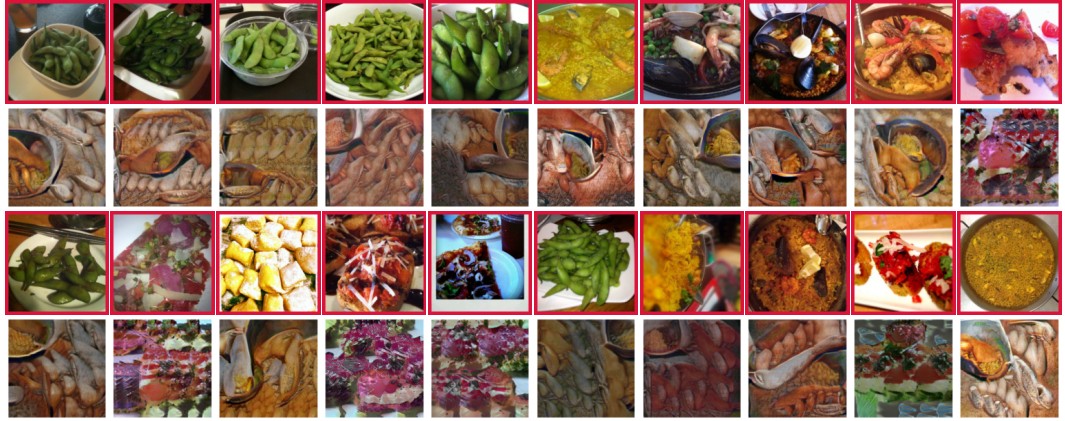

Figure 21: Reconstructions using activation maximization on the input to $\phi$

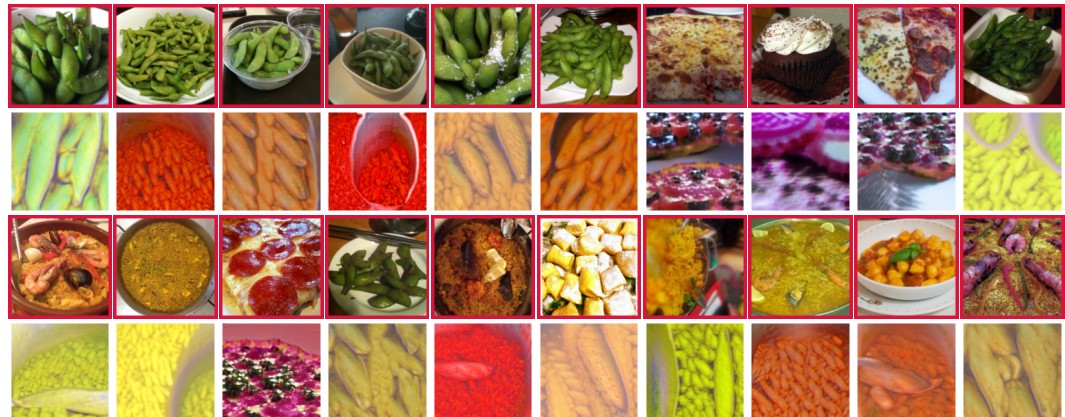

Figure 22: Reconstructions using activation maximization on the input to $\mathcal{F}$

We compare our reconstruction results to a popular baseline for reconstructing data from trained model. It is called "model inversion" in the context of privacy(Fredrikson et al., 2015) or "activation maximization" in the context of visualization techniques (Mahendran & Vedaldi, 2016) (we prefer the term activation maximization as it is more accurate). We are searching for inputs to the model that achieve high activations for the model's outputs that correspond to each class. We consider two options in our case:

The first, by performing activation maximization on the inputs to $\phi$:

$$\underset{\mathbf{x}}{\operatorname{argmin}} \, \mathcal{L}\left(\Phi(\mathbf{x}), \, y\right)$$

This results in multiple candidates $\{\mathbf{x}\}$ that minimize the loss function (binary cross-entropy) w.r.t to the classes $y \in \{-1, 1\}$. We then search for candidates that are nearest neighbours of original training embeddings, and invert them to images by computing $\mathcal{F}^{-1}(\mathbf{x})$ (this is the same pipeline as we use for the reconstructed candidates of our approach). The results of this approach can be seen in Fig. 21.

The second approach, is to optimize over the inputs to $\mathcal{F}$ (instead of the inputs to $\phi$) in the same manner that is described in Appendix B.3:

$$\mathbf{x} = g_\nu(z) \quad s.t. \quad \nu = \underset{\nu}{\operatorname{argmin}} \, \mathcal{L}\left(\phi\left(\mathcal{F}\left(g_\nu(z)\right)\right), y\right)$$

Where $g$ is the U-Net model with parameters $\nu$. This is equivalent to performing the inversion method as described in Appendix B.3, but feeding the output of $\mathcal{F}$ into the trained $\phi$ and then into the loss $\mathcal{L}$, with a given $y \in \{-1, 1\}$ (instead of comparing the output of $\mathcal{F}$ to a given embedding vector). Note that as described in Appendix B.3, the only optimization variables are the parameters of the Deep-Image Prior $g$ (denoted as $\nu$ in the equation above). The results of this approach are shown in Fig. 22.

As evident from the results, while activation maximization techniques manage to reconstruct some interesting outputs, that are somewhat semantically related to the training classes, the results of both methods are inferior to the results of our proposed approach.

## A.11 GT INVERSION

Here are the full results (on all reconstructed candidates) of the results shown in Fig. 9.

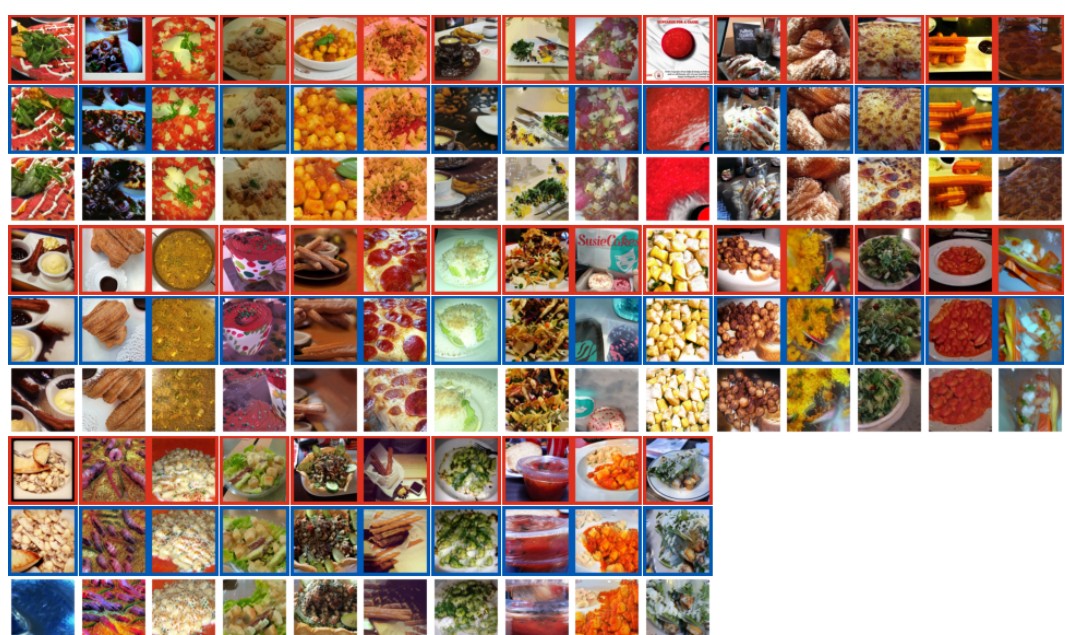

Figure 23: ViT on Food101: Training Image (red), Inversion of Original Embedding (blue) and Inversion of Reconstructed Embedding.

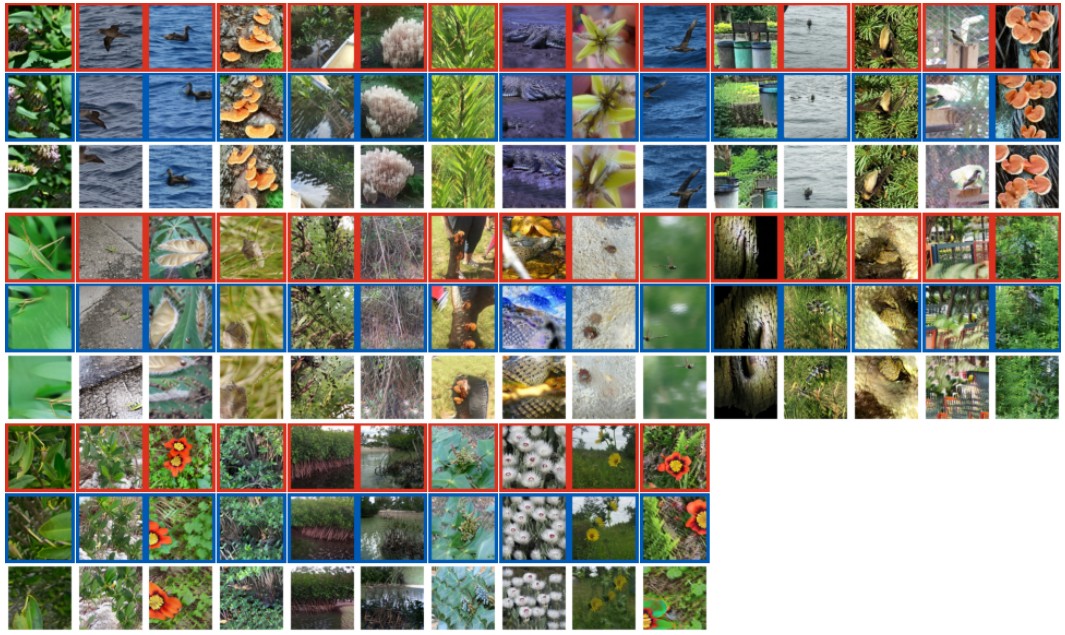

Figure 24: ViT on iNaturalist: Training Image (red), Inversion of Original Embedding (blue) and Inversion of Reconstructed Embedding.

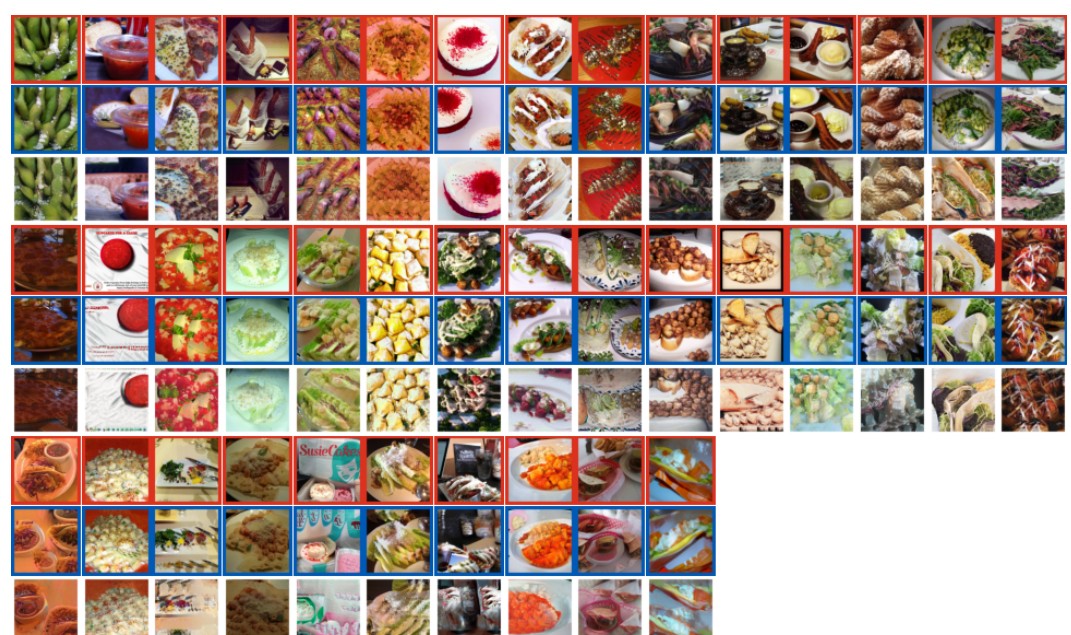

Figure 25: DINO on Food101: Training Image (red), Inversion of Original Embedding (blue) and Inversion of Reconstructed Embedding.

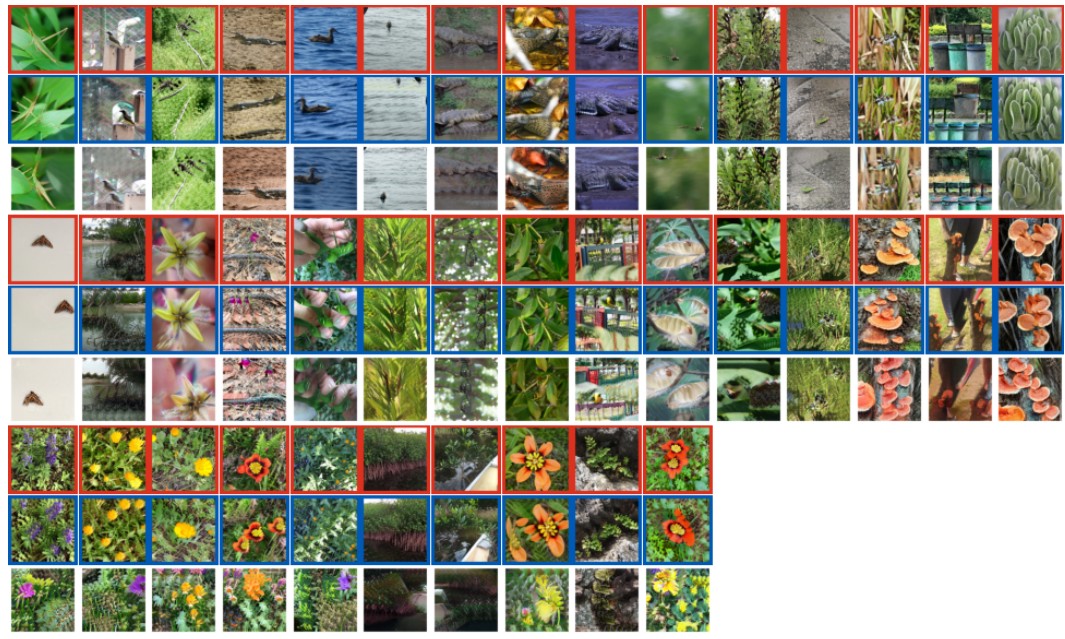

Figure 26: DINO on iNaturalist: Training Image (red), Inversion of Original Embedding (blue) and Inversion of Reconstructed Embedding.

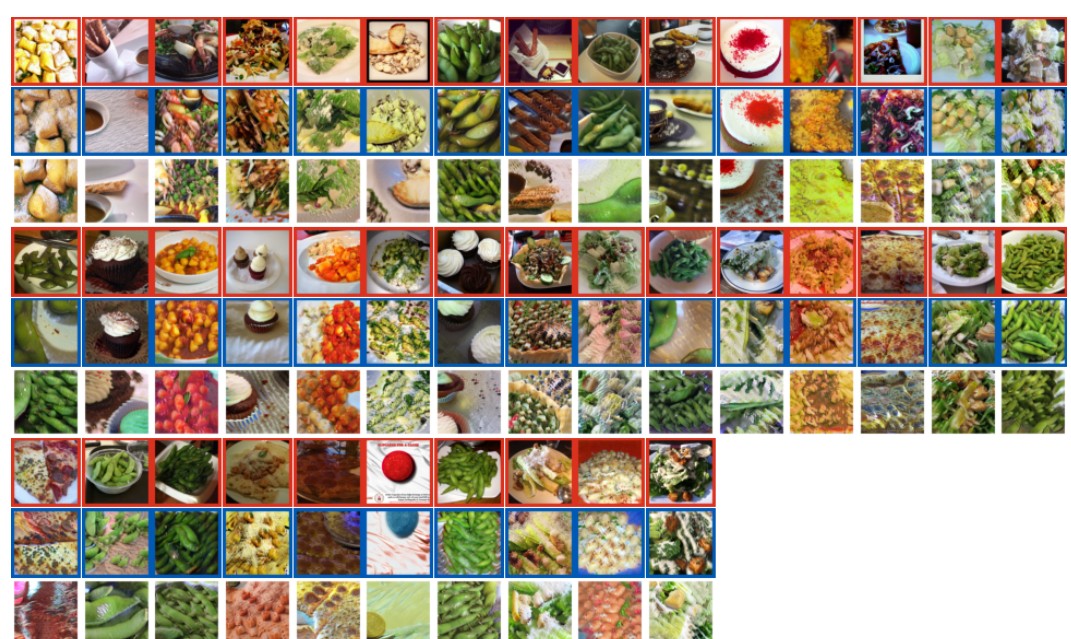

Figure 27: DINO2 on Food101: Training Image (red), Inversion of Original Embedding (blue) and Inversion of Reconstructed Embedding.

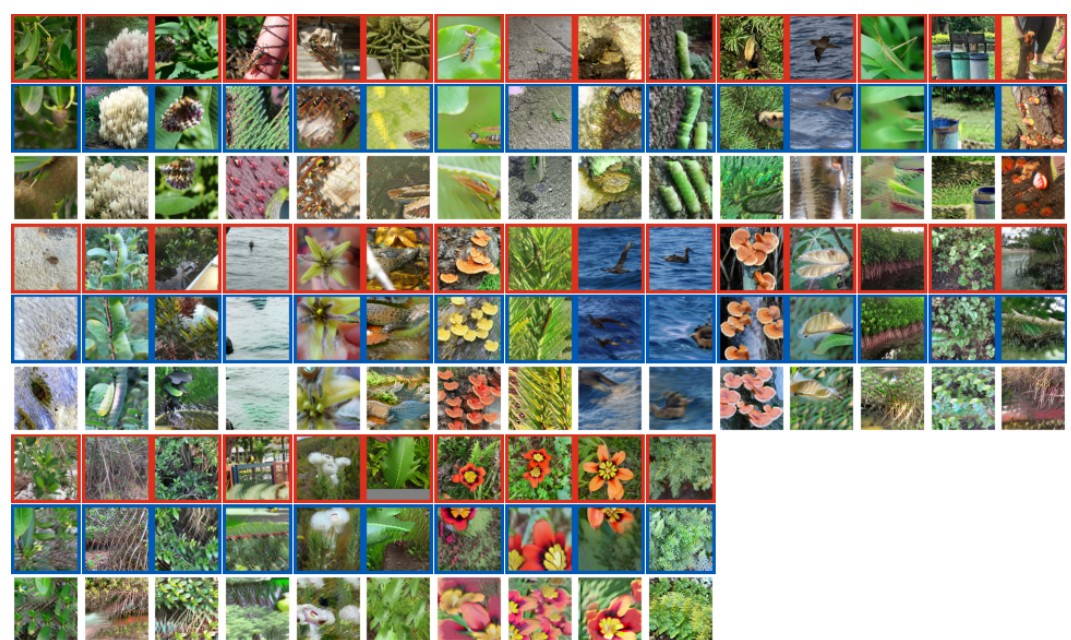

Figure 28: DINO2 on iNaturalist: Training Image (red), Inversion of Original Embedding (blue) and Inversion of Reconstructed Embedding.

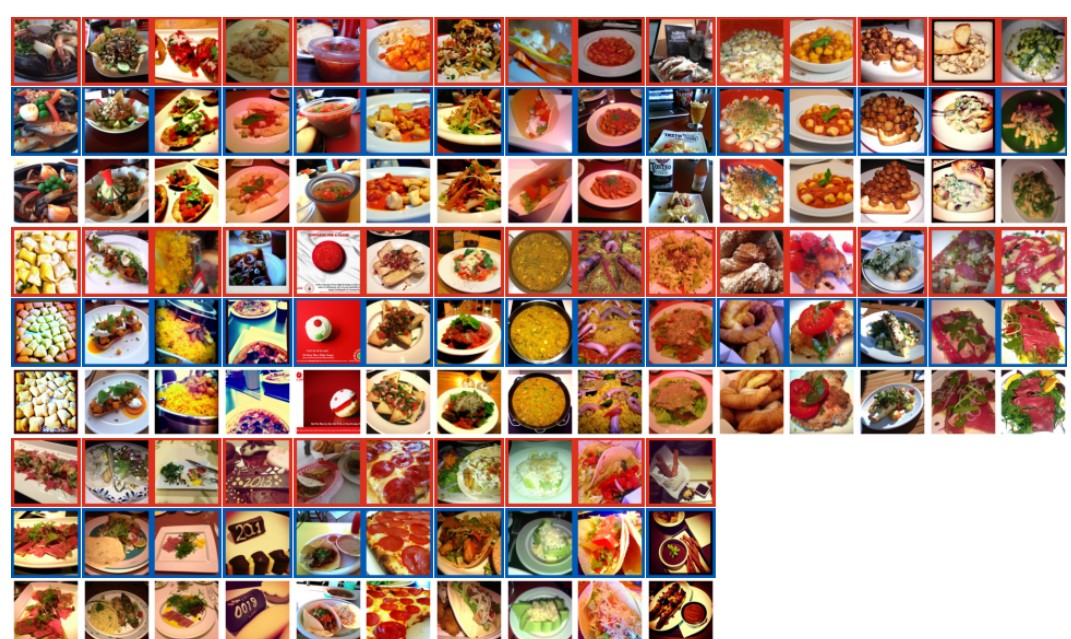

Figure 29: CLIP on Food101: Training Image (red), UnCLIP of Original Embedding (blue) and UnCLIP of Reconstructed Embedding.

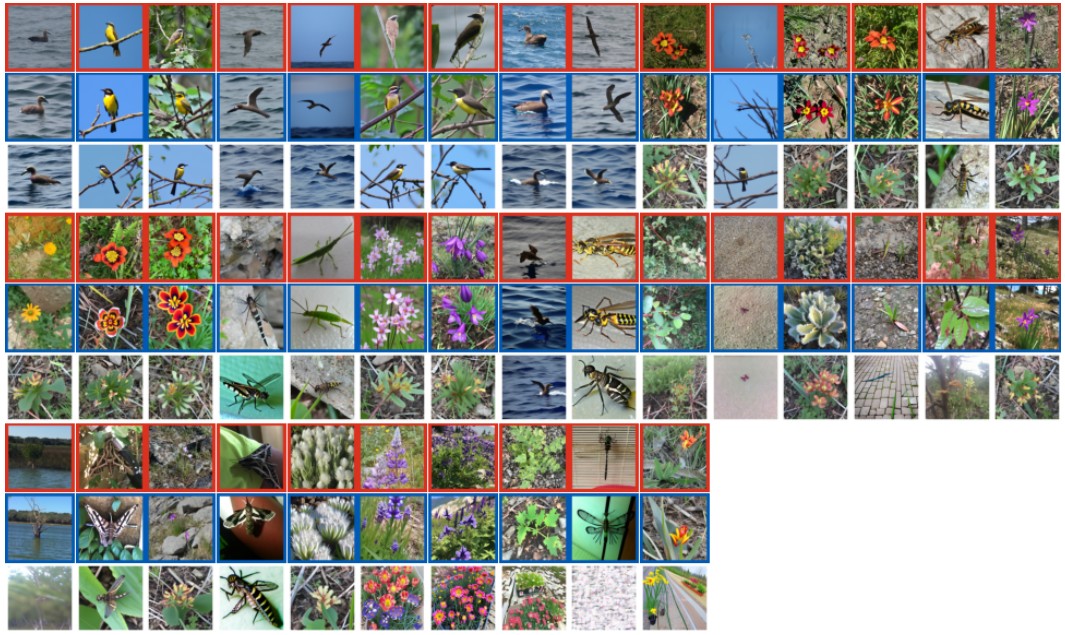

Figure 30: CLIP on iNaturalist: Training Image (red), UnCLIP of Original Embedding (blue) and UnCLIP of Reconstructed Embedding.

## B  Implementation Details

Our code is implemented with PyTorch (Paszke et al., 2019) framework.

### B.1  Data Preprocessing

We resize each image to a resolution of 224 pixels (the smaller side of the image) and then apply a center crop to obtain a $224 \times 224$ image. We then normalize the image per pixel following the normalization used in the original paper of each model, as shown in the table below:

| Model | Mean | Std |
|---|---|---|
| DiNO, DiNOv2 | $[0.485, 0.456, 0.406]$ | $[0.229, 0.224, 0.225]$ |
| ViT | $[0.5, 0.5, 0.5]$ | $[0.5, 0.5, 0.5]$ |
| CLIP | $[0.481, 0.458, 0.408]$ | $[0.269, 0.261, 0.276]$ |

After feeding the images through the backbone $\mathcal{F}$ to obtain the image embeddings $\mathcal{F}(\mathbf{s}_i)$, we normalize each embedding by subtracting the mean-embedding and dividing by the std. Formally: $\mathbf{x}_i = \frac{\mathcal{F}(\mathbf{s}_i) - \mu}{\sigma}$, where $\mu = \frac{1}{n} \sum_{i=1}^{n} \mathcal{F}(\mathbf{s}_i)$, and $\sigma = \sqrt{\frac{1}{n-1} \sum_{i=1}^{n} \left( \mathcal{F}(\mathbf{s}_i) - \mu \right)^2}$.

This is a fairly common approach when training on small datasets. $\mu$ and $\sigma$ can be thought as being part of the model $\phi$ as they are also applied for embeddings from outside the training set.

### B.2  Reconstruction Hyperparameter Search

As mentioned in Section 3.1, we run the reconstruction optimization 100 times with different choice of the 4 hyperparameters of the reconstruction algorithm:

1. Learning rate

2. $\sigma$ – the initial s.t.d. of the initialization of the candidates

3. $\lambda_{\min}$ – together with the loss Eq. (2), the reconstruction includes another loss term to require $\lambda_i > \lambda_{\min}$ (a consequence of the KKT conditions is that $\lambda_i > 0$, but if $\lambda_i = 0$ it has no relevance in the overall results, therefore a minimal value $\lambda_{\min}$ is set.).

4. $\alpha$ – Since the derivative of ReLU is piecewise constant and non-continuous, the backward function in each ReLU layer in the original model is replaced with the derivative of SoftRelu with parameter $\alpha$.

For full explanation of the hyperparameters, please refer to Haim et al. (2022). Note that for $m = 500$, running 100 times would result in 50k candidates.

The hyperparameter search is done via Weights&Biases (Biewald, 2020), with the following randomization (it is in the format of a W&B sweep):

```
parameters:
  random_init_std:
    distribution: log_uniform_values
    max: 1
    min: 1e-06
  optimizer_reconstructions.lr:
    distribution: log_uniform_values
    max: 1
    min: 1e-06
  loss.lambda_regularizer.min_lambda:
    distribution: uniform
    max: 0.5
    min: 0.01
  activation.alpha:
    distribution: uniform
    max: 500
    min: 10
```

### B.3 FURTHER DETAILS ABOUT INVERSION SECTION 3.2

We follow similar methodology to Tumanyan et al. (2022), using their code[9] and changing the reconstruction loss from MSE to Cosine-Similarity as mentioned in Section 3.2, and specifically Eq. (3) (see justifications in Appendix A.1).

The Deep-Image Prior model $g$ is a fully convolutional U-Net model (Ronneberger et al., 2015) (initialized at random with the default pytorch implementation). The optimization is run for 20,000 iterations, where at each iteration the input to $g$ is $z+r$, where $z$ is initialized from $z \sim \mathcal{N}(\mathbf{0}_{d_s}, \mathbb{I}_{d_s \times d_s})$ and kept fixed throughout the optimization, and $r$ is sampled at each iteration as follows:

$$
\begin{aligned}
\text{iteration } i < 10{,}000: \quad & r \sim \mathcal{N}(\mathbf{0}_{d_s}, 10 \cdot \mathbb{I}_{d_s \times d_s}) \\
\text{iteration } 10{,}000 < i \leq 15{,}000: \quad & r \sim \mathcal{N}(\mathbf{0}_{d_s}, \ 2 \cdot \mathbb{I}_{d_s \times d_s}) \\
\text{iteration } 15{,}000 < i \leq 20{,}000: \quad & r \sim \mathcal{N}(\mathbf{0}_{d_s}, 0.5 \cdot \mathbb{I}_{d_s \times d_s})
\end{aligned}
$$

Note that the input to $g$ is of the same size of the input to $\mathcal{F}$, which is simply and image of dimensions $d_s = c \times h \times w$. At each iteration, the output of $g$ is fed to $\mathcal{F}$, and the output of $\mathcal{F}$ (which is an embedding vector of dimension $d = 768$), is compared using cosine-similarity to the embedding vector that we want to invert. At the end of the step, the parameters of $g$ are changed to increase the cosine similarity between the embeddings.

### B.4 INVERSION WITH UNCLIP

While the method in Appendix B.3 is used for ViT, DINO and DINOv2, for CLIP we use a different method to invert, which is by using the UnCLIP implementation of Lee et al. (2022). Unlike the inversion in Appendix B.3 that uses cosine-similarity, with UnCLIP, the embeddings (that go into to UnCLIP decoder) should have the right scale. For each CLIP embedding of a training image ($\mathbf{x}$), we search for its nearest neighbour candidate ($\hat{\mathbf{x}}$) with cosine similarity, but before feeding $\hat{\mathbf{x}}$ into the UnCLIP decoder, we re-scale it to have the same scale as $\mathbf{x}$, so that the input to the decoder is in fact $(\|\mathbf{x}\|/\|\hat{\mathbf{x}}\|)\hat{\mathbf{x}}$. Unfortunately we could not resolve this reliance on the training set (as is also done in previous reconstruction works, and discussed in the main paper), but we believe this may be mitigated by computing and using general statistics of the training set (instead of specific training samples). We leave this direction for future research.

### B.5 RECONSTRUCTION IN MULTICLASS SETUP

The method in Section 3.1 was extended to multiclass settings by Buzaglo et al. (2023). In a nutshell, the reconstruction loss in Eq. (2) contains the gradient (w.r.t. $\boldsymbol{\theta}$) of $y_i \phi(\mathbf{x}_i)$ which is the distance from the decision boundary. For multiclass model $\phi : \mathbb{R}^d \to \mathbb{R}^C$, the distance to the decision boundary is $\phi(\mathbf{x}_i)_{y_i} - \max_{j \neq y_i} \phi(\mathbf{x}_i)_j$ . Replaced into the reconstruction loss in Eq. (2), we have:

$$
L_{\text{rec}}(\hat{\mathbf{x}}_1, \ldots, \hat{\mathbf{x}}_m, \lambda_1, \ldots, \lambda_m) := \left\| \boldsymbol{\theta} - \sum_{i=1}^{m} \lambda_i \nabla_{\boldsymbol{\theta}} \left[ \phi(\hat{\mathbf{x}}_i, \boldsymbol{\theta})_{y_i} - \max_{j \neq y_i} \phi(\hat{\mathbf{x}}_i, \boldsymbol{\theta})_j \right] \right\|_2^2
$$

### B.6 CHOICE OF WEIGHT DECAY

When training our model, we apply weight decay regularization. However, determining the optimal weight decay (WD) value is not straightforward. To find a WD value, we conduct a search across different WD values and observe their impact on test accuracy. The reuslts are shown in Fig. 31. We notice that for most values, the test accuracy increases until approximately 0.1 and then decreases from about 0.3 (indicating that WD is too large). We select the WD from this range, either 0.08 or 0.16. A red-x marks the run which was selected for reconstruction (and whose results are shown in Fig. 3).

---

[9]https://splice-vit.github.io/

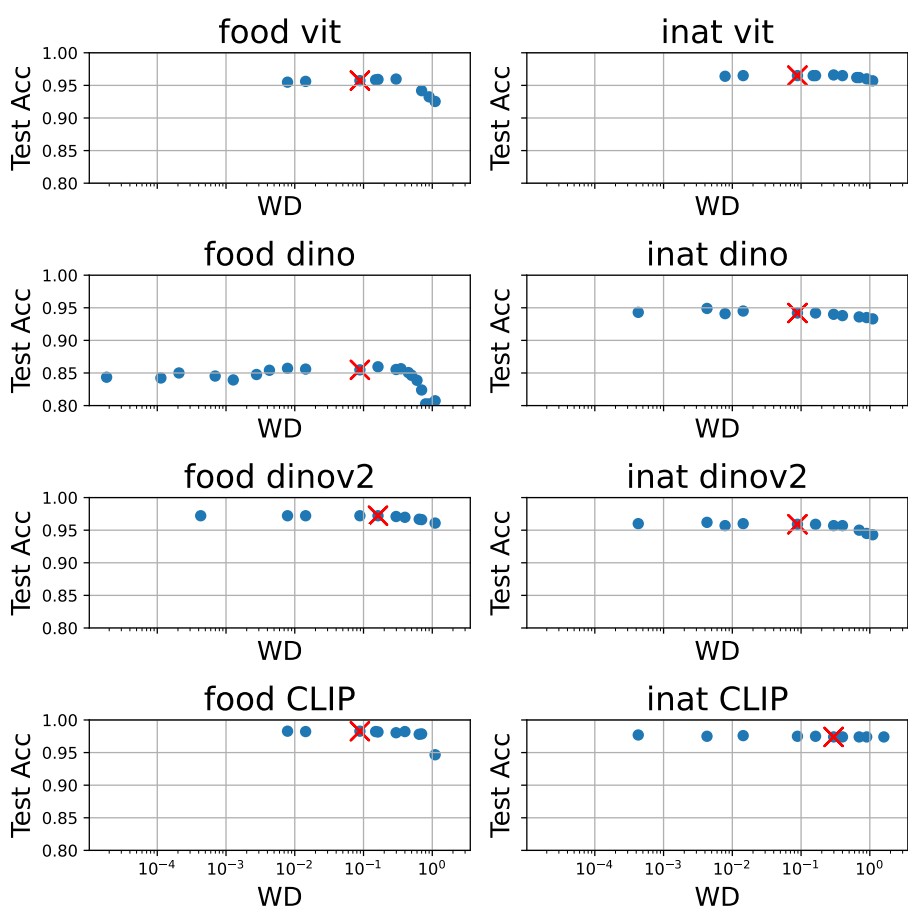

Figure 31: Test-Accuracy for different choices of Weight-Decay Value. Red-X marks the specific run used for reconstruction in Fig. 3

## C  DATASETS - FULL DETAILS

### C.1  IMAGE RESOLUTION

Fig. 32 illustrates how images in the datasets we used may have different resolutions. To standardize the input, we use the pre-processing described in Appendix B.1.

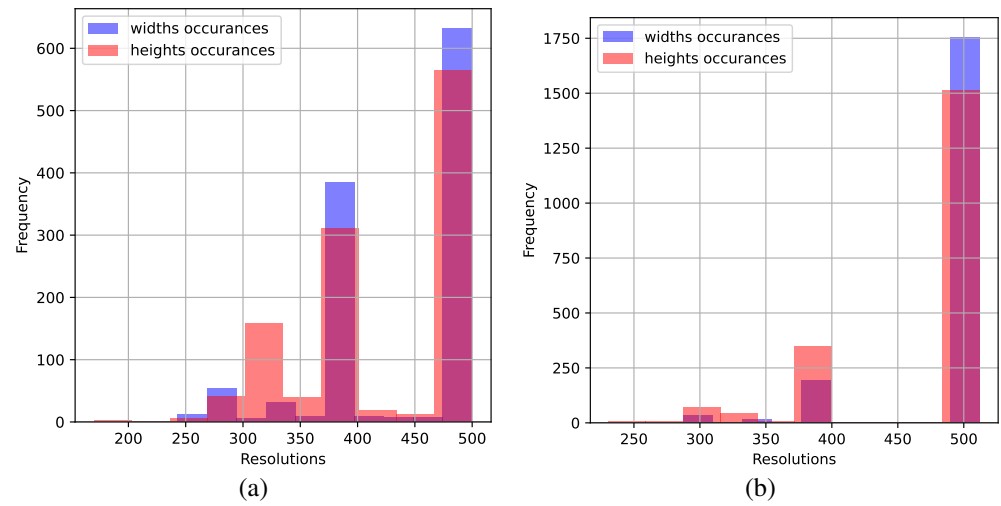

(a)         (b)

Figure 32: Resolution frequency of the images in use from iNaturalist (a) and Food101 (b) datasets

### C.2  FOOD-101 (BOSSARD ET AL., 2014)

The dataset comprises real images of the 101 most popular dishes from the foodspotting website.

**Binary Tasks**  We use the following classes:

- Class I: "beef carpaccio", "bruschetta", "caesar salad", "churros" and "cup cakes"
- Class II: "edamame", "gnocchi", "paella", "pizza" and "tacos"

For any choice of training samples amount, we randomly pick half from every such combined class in order to create our new dataset.

**Multiclass Tasks**  We use the following classes:

"beef carpaccio", "beet salad", "carrot cake", "cup cakes", "dumplings", "gnocchi", "guacamole", "nachos", "pizza" and "samosa"

Here the classes are not combined. For every choice of $N$ classes we choose the first $N$ out of the list above and randomly pick examples according to the training set size and in such a way that the newly formed dataset is balanced.

### C.3  INATURALIST (VAN HORN ET AL., 2018)

The dataset encompasses a total of 10,000 classes, each representing a distinct species.

**Binary Tasks**  Classes are combined in the same manner as for the Food101 dataset. All classes names below appear as they are in the dataset.

**Fauna**

```
02590 Animalia Arthropoda Insecta Odonata Macromiidae Macromia taeniolata
02510 Animalia Arthropoda Insecta Odonata Libellulidae Libellula forensis
```

```
02193_Animalia_Arthropoda_Insecta_Lepidoptera_Sphingidae_Eumorpha_vitis
02194_Animalia_Arthropoda_Insecta_Lepidoptera_Sphingidae_Hemaris_diffinis
00828_Animalia_Arthropoda_Insecta_Hymenoptera_Vespidae_Polistes_chinensis
00617_Animalia_Arthropoda_Insecta_Hemiptera_Pentatomidae_Dolycoris_baccarum
02597_Animalia_Arthropoda_Insecta_Orthoptera_Acrididae_Acrida_cinerea
05361_Animalia_Mollusca_Gastropoda_Stylommatophora_Philomycidae_Megapallifera_mutabilis
04863_Animalia_Chordata_Reptilia_Crocodylia_Crocodylidae_Crocodylus_niloticus
04487_Animalia_Chordata_Aves_Procellariiformes_Diomedeidae_Phoebastria_nigripes
04319_Animalia_Chordata_Aves_Passeriformes_Tyrannidae_Myiozetetes_cayanensis
```

### Flora

```
 05690_Fungi_Basidiomycota_Agaricomycetes_Polyporales_Polyporaceae_Trametes_coccinea
05697_Fungi_Basidiomycota_Agaricomycetes_Russulales_Auriscalpiaceae_Artomyces_pyxidatus
05982_Plantae_Tracheophyta_Liliopsida_Asparagales_Iridaceae_Olsynium_douglasii
05988_Plantae_Tracheophyta_Liliopsida_Asparagales_Iridaceae_Sparaxis_tricolor
06988_Plantae_Tracheophyta_Magnoliopsida_Asterales_Asteraceae_Silphium_laciniatum
06665_Plantae_Tracheophyta_Magnoliopsida_Asterales_Asteraceae_Calendula_arvensis
07032_Plantae_Tracheophyta_Magnoliopsida_Asterales_Asteraceae_Syncarpha_vestita
07999_Plantae_Tracheophyta_Magnoliopsida_Fabales_Fabaceae_Lupinus_arcticus
07863_Plantae_Tracheophyta_Magnoliopsida_Ericales_Primulaceae_Myrsine_australis
08855_Plantae_Tracheophyta_Magnoliopsida_Malpighiales_Rhizophoraceae_Rhizophora_mangle
09143_Plantae_Tracheophyta_Magnoliopsida_Ranunculales_Berberidaceae_Berberis_bealei
09974_Plantae_Tracheophyta_Polypodiopsida_Polypodiales_Pteridaceae_Cryptogramma_acrostichoid
```

## Multiclass Tasks

### 1. Insects

```
 02590_Animalia_Arthropoda_Insecta_Odonata_Macromiidae_Macromia_taeniolata
01947_Animalia_Arthropoda_Insecta_Lepidoptera_Nymphalidae_Phaedyma_columella
02194_Animalia_Arthropoda_Insecta_Lepidoptera_Sphingidae_Hemaris_diffinis
02195_Animalia_Arthropoda_Insecta_Lepidoptera_Sphingidae_Hemaris_fuciformis
02101_Animalia_Arthropoda_Insecta_Lepidoptera_Pieridae_Pontia_occidentalis
02138_Animalia_Arthropoda_Insecta_Lepidoptera_Riodinidae_Apodemia_virgulti
```

### 2. Aquatic Animals

```
 02715_Animalia_Arthropoda_Malacostraca_Decapoda_Grapsidae_Grapsus_grapsus
02850_Animalia_Chordata_Actinopterygii_Perciformes_Lutjanidae_Ocyurus_chrysurus
02799_Animalia_Chordata_Actinopterygii_Perciformes_Centrarchidae_Ambloplites_rupestris
02755_Animalia_Arthropoda_Merostomata_Xiphosurida_Limulidae_Limulus_polyphemus
02704_Animalia_Arthropoda_Malacostraca_Decapoda_Cancridae_Cancer_borealis
02706_Animalia_Arthropoda_Malacostraca_Decapoda_Cancridae_Cancer_productus
```

### 3. Reptiles

```
 04859_Animalia_Chordata_Reptilia_Crocodylia_Alligatoridae_Alligator_mississippiensis
04868_Animalia_Chordata_Reptilia_Squamata_Agamidae_Agama_picticauda
04862_Animalia_Chordata_Reptilia_Crocodylia_Crocodylidae_Crocodylus_moreletii
04865_Animalia_Chordata_Reptilia_Rhynchocephalia_Sphenodontidae_Sphenodon_punctatus
04954_Animalia_Chordata_Reptilia_Squamata_Colubridae_Pituophis_deppei
```

### 4. Birds

```
04487_Animalia_Chordata_Aves_Procellariiformes_Diomedeidae_Phoebastria_nigripes
04319_Animalia_Chordata_Aves_Passeriformes_Tyrannidae_Myiozetetes_cayanensis
04570_Animalia_Chordata_Aves_Suliformes_Phalacrocoracidae_Microcarbo_melanoleucos
04587_Animalia_Chordata_Aves_Suliformes_Sulidae_Sula_nebouxii
04561_Animalia_Chordata_Aves_Strigiformes_Strigidae_Surnia_ulula
04576_Animalia_Chordata_Aves_Suliformes_Phalacrocoracidae_Phalacrocorax_capensis
```

