# SUPPLEMENTARY MATERIAL FOR SUBMISSION:

# RECONSTRUCTING TRAINING DATA FROM REAL WORLD MODELS TRAINED WITH TRANSFER LEARNING

## 1 FULL RESULTS FOR FIG.5B FROM MAIN PAPER

In Figure 5b in the main paper we show reconstructed samples, sorted according to their reconstruction-quality as measured by cosine-similarity between their image embeddings. The results shown there are obtained from a model trained on DINO embeddings on Food101 dataset.

Below we provide the complete results for this type of evaluation, namely, for each model from Figure 3, and for each reconstruction-quality metric, we show the "best" reconstructed samples according to this metric (by sorting them).

In total there are 8 models: trained on 4 backbones (ViT, DINO, DINOv2 and CLIP) and on 2 datasets (Food101 and iNaturalist), as described in details in the Results Section in the main paper. And for each model we show the sorted results for a total of 6 choices for reconstruction-quality: Cosine-Similarity in Embedding space plus 5 metrics in Image space: SSIM, LPIPS, Split-Product, PSNR and Cosine-Similarity (Image space).

In each Figure below, images with RED borderline are original training images, and the image below them is their nearest reconstructed image (as measured by the cosine-similarity between their embeddings). Note: in all cases, the matching between a training image and its reconstructed image is the same. the only difference is between the way they are sorted – which is done using the metric (as written in the left side of each row).

We are aware that there may be some sampling bias in these results. However, since in our work, the inversion part is time-intensive, we must choose which embeddings to invert, where in our work we use the cosine-similarity for that, as detailed in the paper. It is not feasible to use an image-metric for this goal, because this would mean first inverting all candidate embeddings (usually 25k-50k of them), which is not feasible.

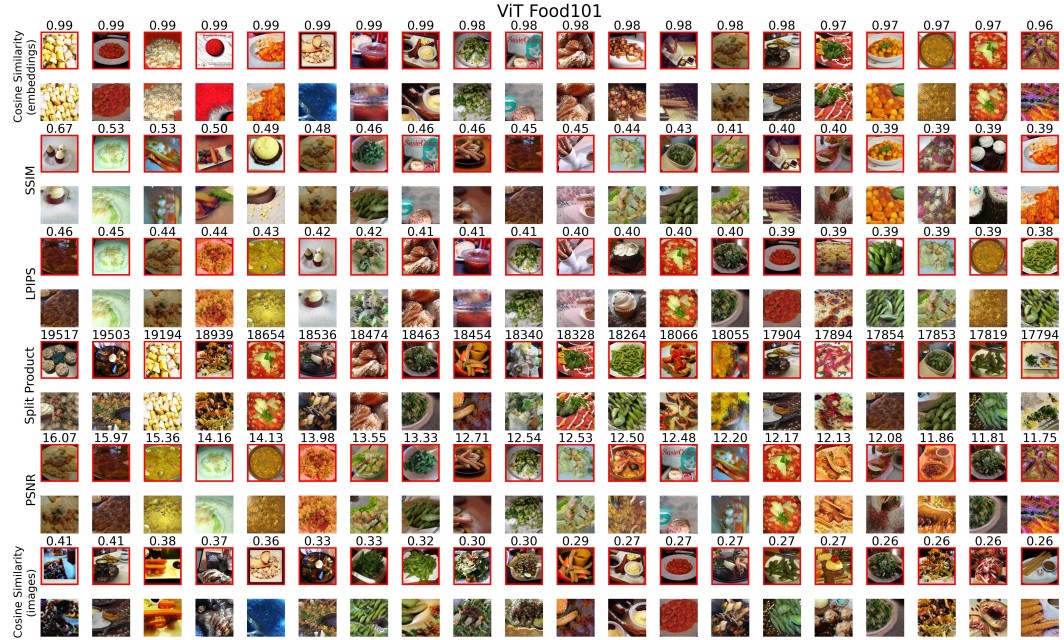

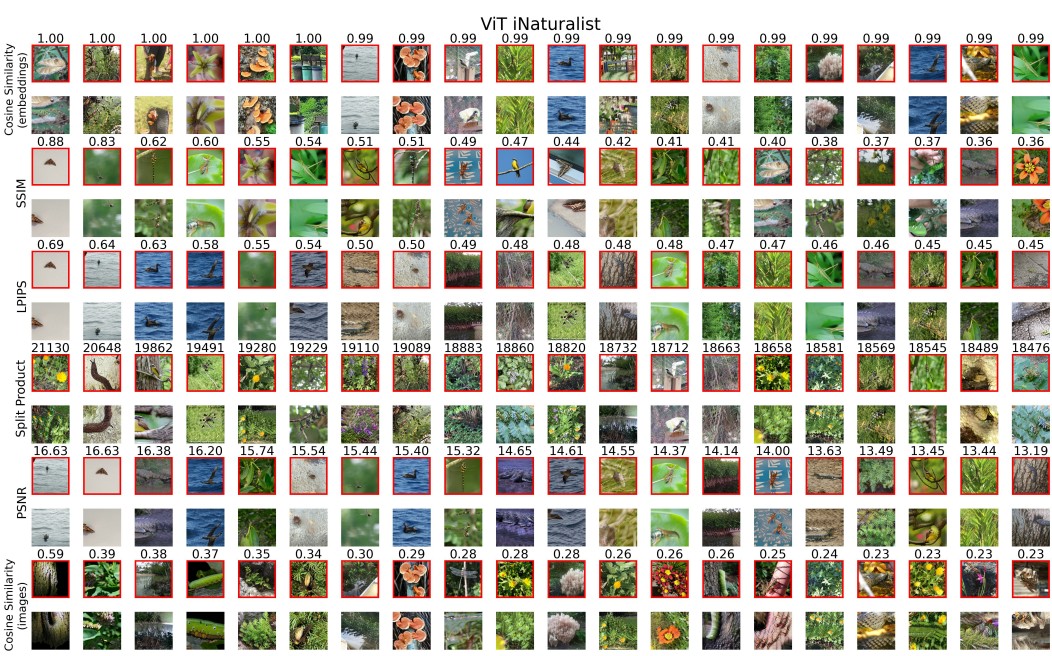

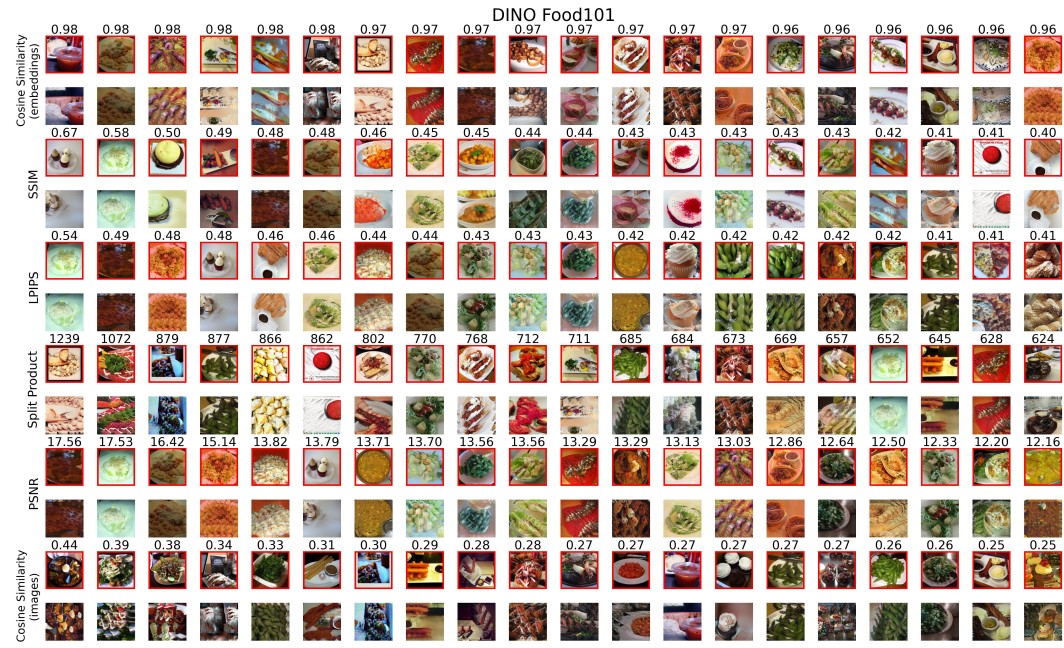

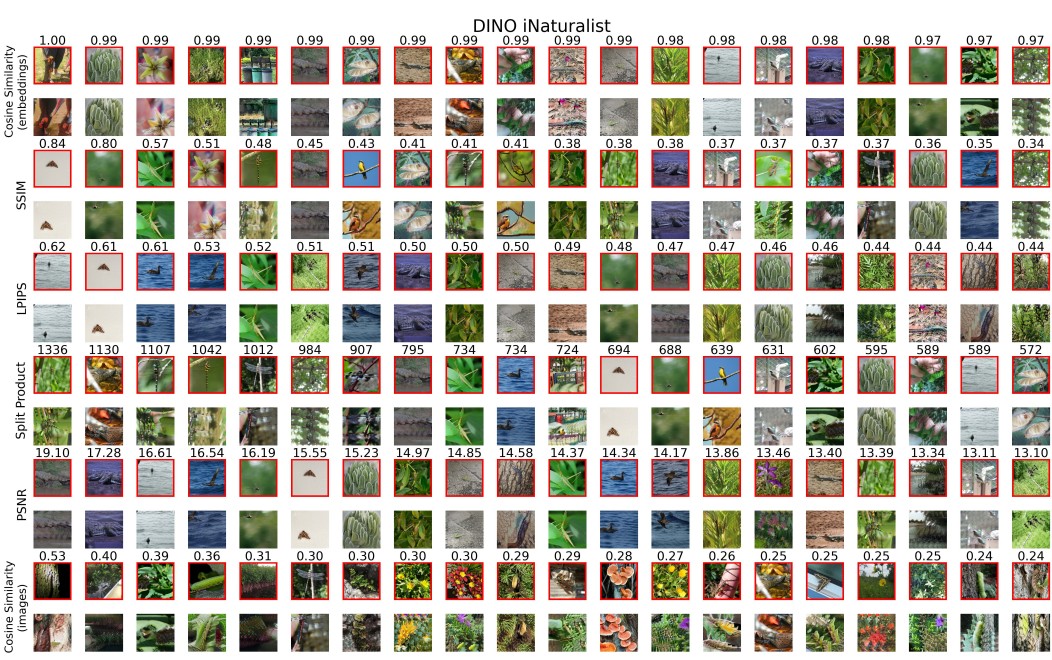

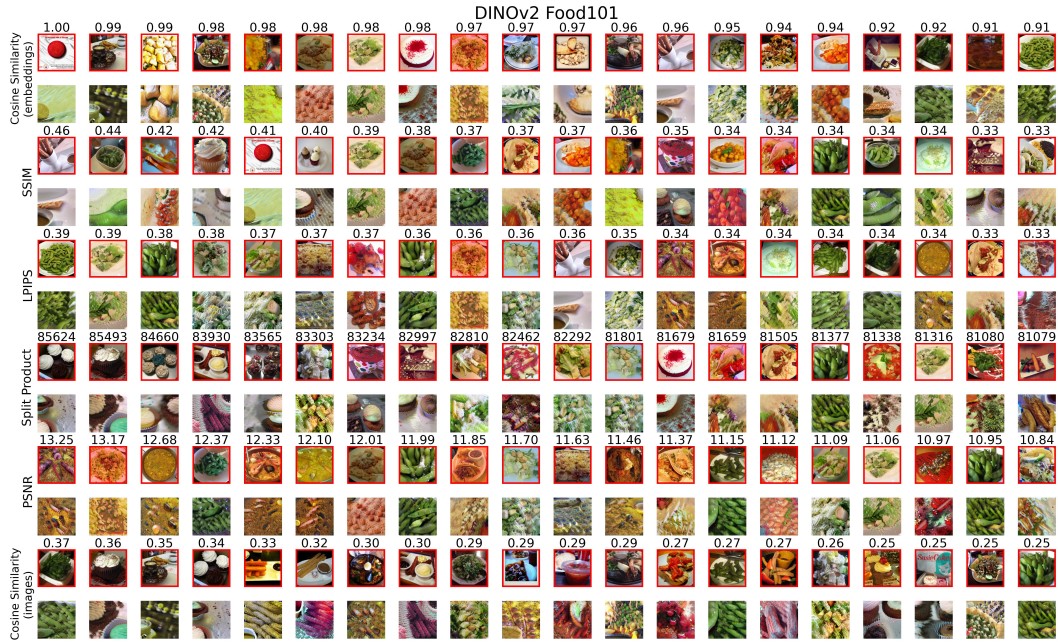

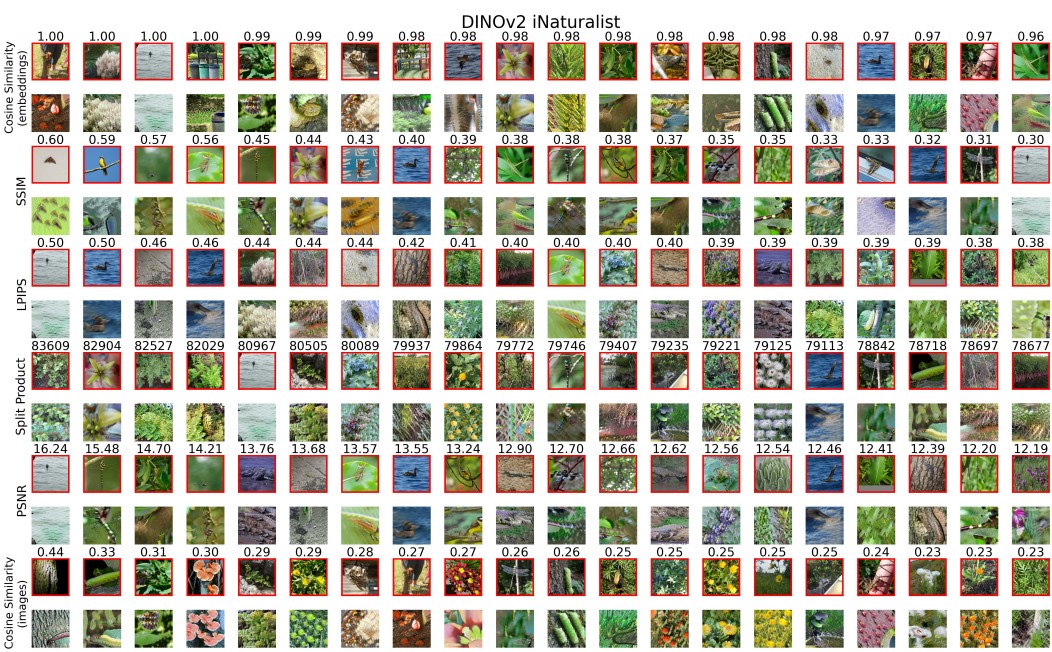

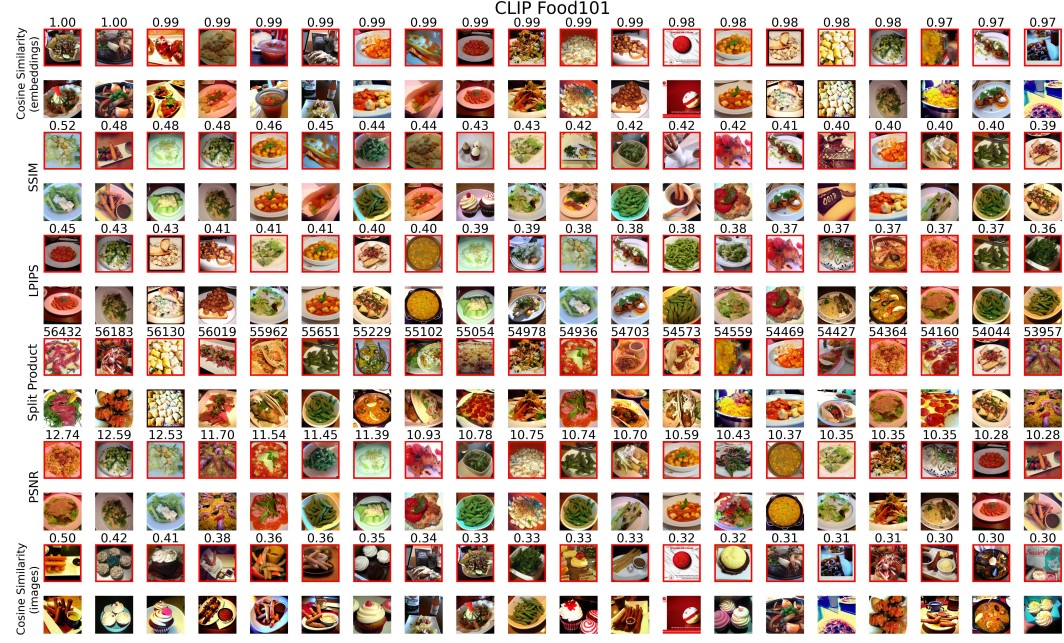

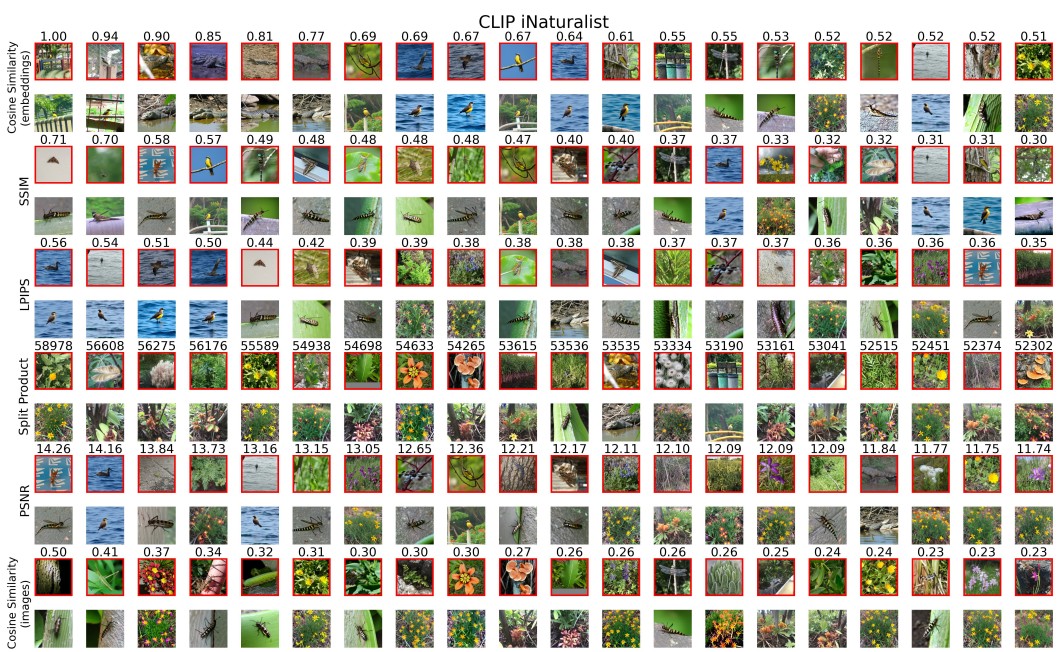