# OpenReview forum: "Reconstructing Training Data From Real-World Models Trained with Transfer Learning"
_ICLR.cc/2025/Conference — Submitted to ICLR 2025_

### Official Review · Reviewer_VVGm · 2024-10-28

**Soundness:** 2
**Presentation:** 2
**Contribution:** 2
**Rating:** 5
**Confidence:** 5

**Summary:**

This paper proposes a new method for training data reconstruction. Different from other works, this paper proposes to recover the image embedding first, and then employs a inversion network to reconstruct the images. Especially, this paper proposes to use the clustering-based approach for effectively identifying training samples. Experiments show the performance of the proposed method.

**Strengths:**

The proposed method has show better performance compared with baselines, on the datasets of Food-101 and iNaturalist.

**Weaknesses:**

1. The experiments are only conducted on the CLIP and DINO which has served as the common components for the image diffusion models. However, I think the proposed network should be suitable for various networks. I wonder the training data reconstruction with other basic networks, like ResNet. And the authors should explain why they focused on transformer-based models in the experimental section.

2. This paper claims to be suitable for the reconstruction of high-resolution images, while the experiments are only conducted on the datasets with low resolution like 224 (why 224x224 fits that definition of "high resolution" in this paper?). I wonder the performance with higher resolution like 512.

3. There is no quantitative metric to compare the reconstruction effects between the proposed method and baselines. For example, using the reconstructed images can lead to a model with the similar accuracy of the original model?

4. The presentation of this paper is not good. For example, the texts in Fig.9 has be out of the width constraint, which could be resized.

**Questions:**

1. What is the performance of the proposed method with more types of networks besides the transformer?

2. What is the reconstruction performance on images with higher resolution?

3. Can the authors provide more reliable quantitative metrics?

**Details Of Ethics Concerns:**

The proposed method can be utilized to steal the training data of trained models, influencing the privacy requirement.

---

> ### Author Response · Authors · 2024-11-19
>
> We thank the reviewer for the thorough feedback.
>
> Answers for “Weaknesses” part:
>
> - This is a good point. Previous works were successful in reconstructing training data only from very small-scale networks (MLPs and CNNs) and small-scale datasets (MNIST and CIFAR), see lines 33-36. In this work, we focused on popular backbone models for transfer learning. As we report in lines 494-497, we attempted our method on larger CNNs (e.g., VGG), and it wasn’t as successful. Reconstructing data from Transformer-based backbones proved more successful. In this respect, it is important to note that transformer-based backbones are very popular for feature extraction for transfer learning tasks in practice, and we believe that reconstructing data from transformer-based feature extractors is an important contribution on its own. We emphasize that our work is the first to reconstruct training images from real-world networks, rather than small-scale and unrealistic models as in previous works. We agree that exploring data reconstruction from other backbones is an important research direction, but beyond the scope of this paper.
>
> - Thank you for pointing this out. Please note that all previous works on data reconstruction from trained models were done on much lower-resolution datasets such as MNIST (28x28) or CIFAR (32x32). Our work is the first to reconstruct training images from higher-resolution images (224x224), comparable to Imagenet-size images. Additionally, note that most standard backbones (like the ones we use), are trained at this resolution, and that typically, input images are first resized (and center-cropped) to this same resolution (224 pixels). Thus, this is a standard resolution. We agree that data reconstruction from higher resolution models is important, but this is beyond the scope of our paper.
>
> - We make a great effort to establish a quantitative method for comparing reconstruction quality. We kindly refer the reviewer to the “Quantitative Evaluation” paragraph in lines 346-400. In particular, we compare the reconstruction quality between 6 different metrics (Figure 5). We additionally compare to activation maximization method (Appendix A.10) which is applicable in our setting. We emphasize that since this is the first work to successfully reconstruct training data from higher resolution images than CIFAR, it is difficult to compare to other methods, as there are none besides activation maximization. The reviewer’s suggestion is very good, we attempted this but it wasn’t successful. A possible reason may be, that quite surprisingly, the output of the network on reconstructed images can be very different from network output on corresponding original images, even though they are visually similar. There seems to be an inherent scaling issue in the reconstruction optimization. It is an interesting direction for future research.
>
> - Thank you for your comment. We will resize Figure 9 to fit the width constraints. If there are other issues with the presentation we would be happy to fix them.
>
> Questions:
>
> - The KKT-based reconstruction is known to be successful for different types of networks (see [Haim et al. 2022], [Buzaglo et al. 2023]). Our method works for networks that can be “inverted” similarly to CLIP/DINO-ViT/ViT, hence we currently don’t know how to extend it directly to ResNets or other types of networks. This is an active field of research that we also work on, although it seems a more difficult task.
>
> - As we mentioned, although it would be interesting to test even higher-resolution images, this is currently beyond the scope of our paper. The resolution 224X224 is used in most standard backbones (like the ones we use), and it is the Imagenet resolution. We think it is surprising by itself that it is possible to reconstruct images beyond a resolution of 32x32, which is the highest resolution achieved by previous works in the field. (also see our answer to weaknesses (2)).
>
> - Please see our answer to weakness (3).

---

> > ### Comment · Reviewer_VVGm · 2024-11-27
> > **Review comments**
> >
> > After reading the response from the authors, some concerns have not be addressed, including the experiments on more types of backbones, experiments with higher resolutions. Therefore, I think this paper still needs to make further explorations, making the contribution of this paper more solid. I will keep my original rating.

---

> > > ### Author Response · Authors · 2024-12-01
> > >
> > > Thank you for your response. The paper already shows results for ViT, Dino-ViT, Dino-ViT2 and CLIP which are extremely popular backbones for transfer learning, and with the respective standard resolution for these backbones. We would appreciate if the reviewer elaborate on the type of experiments that are required.

---

### Official Review · Reviewer_c8yx · 2024-10-29

**Soundness:** 3
**Presentation:** 3
**Contribution:** 2
**Rating:** 5
**Confidence:** 2

**Summary:**

This paper demonstrates the reconstruction of high-resolution training images from models trained using a transfer learning approach, as well as the reconstruction of non-visual data. Moreover, it introduces a novel clustering-based approach for effectively identifying training samples without prior knowledge of the training images.

**Strengths:**

1. The writing is well done and clear.

2. They present the weaknesses and limitations of their method in detail.

3. The experiments consider various commonly used pre-trained feature extractors such as CLIP, demonstrating the effectiveness of their method.

**Weaknesses:**

1. The method is limited to specific cases where a fixed feature extractor and some MLP layers serve as the classifier. It cannot generalize to other more common transfer learning scenarios, such as fine-tuning an entire classifier or certain layers of a classifier.

2. The introduced method lacks innovation. Specifically, the core contributions of reconstructing embedding vectors in Section 3.1 and mapping embedding vectors in Section 3.2 to the image domain either originate from other works or involve only simple modifications, such as changing the MSE loss to the cosine similarity loss. Please clearly explain the novel aspects of the method introduced.

3. The format of Figure 9 is incorrect as it exceeds the page boundary.

**Questions:**

My confusion is the aforementioned weaknesses. If there are any misunderstandings, please point them out.

---

> ### Author Response · Authors · 2024-11-19
>
> We thank the reviewer for the feedback.
>
> - Thank you for pointing this out. In this work, we focus on models trained on embeddings of common backbone models, where the backbone models’ parameters are fixed (the so-called “feature-extractor” approach for transfer learning). We believe that this approach, while limited, is still the most common training approach used in practice. To support this, please see Kim et al. [1] who analyzed multiple papers on transfer learning for medical data and found that using backbones as feature-extractors is the most popular approach for transfer learning. We agree with the reviewer that exploring data reconstruction from models trained with other transfer learning approaches (e.g., fine-tuning backbone parameters) is of great importance and value to the community, however it is beyond the scope of this paper. Whether the current approach can be generalized to such settings is an interesting direction for future works.
>
> - The main contribution is in demonstrating the vulnerability to training-data reconstruction of models that are trained in a transfer learning manner using embeddings of widely-used backbone models. This is a very popular approach for training classifiers on tasks with limited data, and exposing this vulnerability should be of major concern to practitioners. We provide an abundance of experimental evidence to support our claims. Another major contribution is the clustering-based reconstruction approach, which is very important in making such reconstruction attacks useful in practical settings. Note that many works in the field are using the training data in order to validate their attack. Our clustering-based approach alleviates this requirement.
>
> - Right. We will fix it in the final version. Thanks.
>
> [1] “Transfer learning for medical image classification: a literature review”, Kim et al. 2022

---

> > ### Comment · Reviewer_c8yx · 2024-11-24
> >
> > Thank you for author‘s response. Considering author‘s feedback and the limitations of the paper, I will maintain my score.

---

### Official Review · Reviewer_hzzB · 2024-11-04

**Soundness:** 2
**Presentation:** 2
**Contribution:** 1
**Rating:** 3
**Confidence:** 3

**Summary:**

This paper studies the training data reconstruction problem. Different from previous methods, this paper studies the data reconstruction from models trained in a transfer learning approach, and claims the first approach to reconstruct images from the latent features. The experimental section showcases comprehensive results to verify the efficacy of proposed method in different datasets and backbone networks.

**Strengths:**

1. This paper is easy to follow.
2. I appreciate the comprehensive experiments performed to analyze the effectiveness of the proposed approach.

**Weaknesses:**

1. The technique contribution is limited. This paper aims to reconstruct images from latent features (embeddings), however, the key components used for this purpose are borrowed from previous works. Specifically, it use ‘’Reconstructing training data from trained neural networks’’ to reconstruct embeddings, followed by ‘’Splicing vit features for semantic appearance transfer’’ to convert embeddings into RGB images.
2. The resolution of reconstructed images are still low (224x224).
3. The experiment part contains results of this method, but without any comparison with other approaches.

**Questions:**

See weakness

---

> ### Author Response · Authors · 2024-11-19
>
> We thank the reviewer for the feedback.
>
> - The main contribution is in demonstrating the vulnerability to training-data reconstruction of models that are trained in a transfer learning manner using embeddings of widely-used backbone models. This is a very popular approach for training classifiers on tasks with limited data, and exposing this vulnerability should be of major concern to practitioners. We provide an abundance of experimental evidence to support our claims. Another major contribution is the clustering-based reconstruction approach, which is very important in making such reconstruction attacks useful in practical settings. Note that many works in the field are using the training data in order to validate their attack. Our clustering-based approach alleviates this requirement.
>
> - Thank you for pointing this out. Please note that all previous works on data reconstruction from trained models were done on much lower-resolution datasets such as MNIST (28x28) or CIFAR (32x32). Our work is the first to reconstruct training images from higher-resolution images (224x224), comparable to Imagenet-size images. Moreover, note that most standard backbones (like the ones we use) are trained at this resolution, and that typically, input images are first resized (and center-cropped) to this same resolution (224 pixels). Thus, this is a standard resolution. We agree that data reconstruction from higher-resolution models is important, but this is beyond the scope of our paper.
>
> - We compare our method to two activation maximization methods (Appendix A.10) which are applicable in our setting. Since this is the first work to successfully reconstruct training data from higher resolution images than CIFAR, there are no other existing methods that we are aware of, and designed for direct comparison in such settings.

---

### Official Review · Reviewer_qRiu · 2024-11-09

**Soundness:** 4
**Presentation:** 3
**Contribution:** 3
**Rating:** 8
**Confidence:** 4

**Summary:**

This paper explores optimization-based **data inversion** techniques from pre-trained models with transfer learning, in which we could reconstruct the training data simply given the pre-trained encoder and classifier itself. It extends the model-inversion techniques with improved designs like loss, generative decoder. The clustering-based approach further demonstrates the possibility of high-quality reconstructions to the original data. Experiments on two common datasets reveals the potential privacy risks associated with models trained on sensitive data.

**Strengths:**

[**Novelty**]
- Working in embedding space instead of directly reconstructing images is an innovative approach, and makes it more scalable to different models on both visual and non-visual data
- the adoption of clustering and averaging technique is novel, which identifies high-quality reconstructions when training data is not available

[**Significance**]
- it highlights the significant privacy risks associated with models trained with sensitive data in a transfer learning setup, in which high-resolution data reconstruction in real-world conditions emphasizes the critical need for privacy-preserving mechanisms with today's pre-trained models.

[**Completeness & Clarity**]
- High-quality visualizations, including comparisons of reconstructed images to original data and plots that illustrate reconstruction metrics, add depth to the evaluation, making it easier to interpret the results.
- The writing is clear, which effectively lays out the motivation and approach, and well explains the limitations

**Weaknesses:**

- On significance, while I like the simple paradigm of solving x when f is known within f(x)=y, I kind feel that reconstructed data samples are not exactly matching with the actual training data, especially when the training data is not available. It is more close to averaged per-category data when training data is not available, the author might want to turn down their scope.
- Another thinking is that the current method may only work with classifiers-based model, for more more fine-grained training data reconstruction from segmentation model like SAM might be more desired. Also the reconstruction quality varies significantly with the choice of backbone model (DINO, CLIP, etc.), which affects the novelty of the clustering approach by making it model-dependent.
- As the author also mentioned, the inversion process used to map embeddings back to images is computationally expensive, which could hinder scalability

**Questions:**

- I am also wondering whether we have any baselines in this line of data inversion techniques
- a quick question, for diffusion-based generative models, wondering whether it is also a more realistic concern to reveal training data directly.

---

> ### Author Response · Authors · 2024-11-19
>
> We thank the reviewer for the thorough and positive feedback.
>
> Weaknesses:
>
> - While we agree that some results look similar to class average, many are very similar to the original image itself (Fig. 3). We also think that such a comparison should be with respect to other techniques. In appendix A.10 we compare our results with two activation-maximization baselines on the same models. Note that the results from these techniques are far from being similar to any of the original images or their class representatives.
>
> - Indeed, the choice of the backbone is significant to the success of the reconstruction, and especially for this reason we focused our experiments on very popular backbones for vision tasks (DINO-ViT, ViT and CLIP).
>
> - The reviewer is correct in noticing that the inversion process is a bottleneck of our method. However, our approach is independent of
> the specific choice of the inversion technique, so that better inversion methods will certainly achieve better results. Additionally, encoder-based inversion techniques that we use for CLIP, are not so computationally expensive as the optimization-based techniques.
>
> Questions:
>
> - We compared our reconstruction method to two different activation maximization reconstruction schemes. The results are in Appendix A.10 and are significantly inferior to our reconstruction scheme. We additionally compared six different metrics to evaluate the reconstruction results (Figure 5), and in Figure 6, we show a high correlation between an image being close to the margin and having a high reconstruction score, which is predicted by the theory. Given that this is the first work to reconstruct images with a resolution higher than CIFAR from trained classifiers, activation maximization remains the primary method available for comparison in this context.
>
> - Regarding diffusion-based generative models, works that focus on data reconstruction from such modes (e.g., Carlini et al. 2023) have a very different approach. They are also highly dependent on knowing the training data for their evaluation (e.g., specifically targeting highly duplicated samples in the training set, that are searched for prior to the attack, and then using their text prompts from the training data in order to generate image samples). While such works are of prime importance, our submission focuses on classifier models and emphasizes privacy risks in transfer learning tasks, which are widely used in practice.

---

> > ### Comment · Reviewer_qRiu · 2024-11-26
> >
> > Thanks for the reply, I think my major concerns are well addressed, except for the part on **"more fine-grained training data reconstruction from segmentation model like SAM"**. I also checked all previous papers on training data reconstruction like [Deconstructing Data Reconstruction](https://arxiv.org/abs/2307.01827), and [Reconstructing Training Data from Trained
> > Neural Networks](https://arxiv.org/pdf/2206.07758), the improvements proposed in this paper largely boost the images quality, which demonstrates immediate potential of data reconstruction at the original quality.  I would like to call out that, compared to all previous works, the improvement here is significant. Thus I'm leaning towards slightly increasing my score after the discussion.

---

> > > ### Author Response · Authors · 2024-12-01
> > >
> > > We thank the reviewer for their openness to increasing the score and we highly appreciate their efforts in reviewing prior literature. Regarding reconstruction from segmentation models: [Buzaglo et al. (2023)](https://arxiv.org/abs/2307.01827) showed that data reconstruction may be possible even for models trained with other losses (not limited to classification) with weight decay term. However, the feasibility of such attacks in the context of segmentation models requires extensive investigation. We believe it would be beneficial to begin by testing these tasks on small-scale models before progressing to large-scale models like SAM, even in the context of transfer learning.

---

### Author Response · Authors · 2024-11-24
**Response to all the reviewers**

Again, we thank the reviewers for their comments. We hope that our response has addressed all the issues raised by the reviewers and that they will consider updating their scores accordingly. If there are remaining questions, we would of course be happy to address those before the public discussion phase ends.

Many thanks, The authors

---

### Meta-Review · Area_Chair_umNp · 2024-12-16

**Metareview:**

The paper conducts optimization-based data inversion from pre-trained models with transfer learning. The objective is to reconstruct the training data given the pre-trained encoder and classifiers using clustering techniques. The main contribution is in demonstrating the vulnerability to training-data reconstruction of models trained in a transfer learning manner using embeddings of widely-used backbone models.

Strengths:
- Operates in the embedding space, making it more scalable.
- The use of clustering is novel.
- Highlights privacy risks in models trained with sensitive data.
- Contains comprehensive experiments.

Weaknesses:
- The data reconstructed is the expected data rather than the original data, thus the scope should be adjusted.
- Limited usability due to the required encoder/classifier setup and limited verification on CLIP and DINO.
- Limited technical contribution as it uses existing techniques.

Given the limited technical contribution, the empirical contribution must be substantial and the discussions insightful to make them the main empirical contribution of the paper.  However, the paper has a limited empirical contribution and requires major experiments to fully addressed the concerns raised by the reviewers.  Thus, I recommend the rejection of the paper.

**Additional Comments On Reviewer Discussion:**

Reviewer qRiu recommended the acceptance of the paper based on the fact that the method works on the embedding space instead of the image space and deems this novel. However, this is a common approach in the field and is not a new contribution. Moreover, the reviewer refers to the privacy risks highlighted in the paper. The weaknesses the reviewer cites include toning down the scope and the limited applicability of the proposal, as dense models may not work with it. After the rebuttal, the reviewer mentions that their major concerns were addressed except for evaluations on fine-grained problems such as segmentation.

Reviewer hzzB mentioned that the contribution is limited since the key components used in the proposal were taken from previous work and noted that the resolution (224x224) is still low. The authors replied by emphasizing their main contributions and stated that data reconstruction models typically work with smaller resolutions. The reviewer did not respond to the authors.

Reviewer c8yx mentioned that the method is limited to specific cases and cannot generalize to common transfer learning scenarios. The reviewer also highlighted that the method is not innovative since it relies on existing works. The authors replied to the reviewer by stating that their setup is common and that the main contribution is the demonstration of the vulnerability to training-data reconstruction. The reviewer acknowledged the review but maintained a borderline reject score.

Reviewer VVGm noted that the experiments were only evaluated on CLIP and DINO and that other networks should have been evaluated as well. Similarly to reviewer hzzB, this reviewer mentioned that 224x224 shouldn’t be considered as high resolution. After the rebuttal from the authors, the reviewer still complained about the missing experiments on different setups and considered that the paper needs more exploration to be more solid, maintaining the borderline reject score.

The reviewers did not respond to my post-rebuttal discussion.

Overall, the paper addresses a significant problem, but the results are not significant enough as mentioned by reviewers c8yx and VVGm. Moreover, given the limited technical contribution, the empirical contribution must be substantial and the discussions insightful to make them the main empirical contribution of the paper. This is not the case, as noted by the three negative reviews. The comments from the positive reviewer qRiu are not enough to outweigh these drawbacks. Thus, I recommend the rejection of the paper.

---

### Decision · Program_Chairs · 2025-01-22

Reject